# Modelling of Neuronal Ceroid Lipofuscinosis Type 2 in *Dictyostelium discoideum* Suggests That Cytopathological Outcomes Result from Altered TOR Signalling

**DOI:** 10.3390/cells8050469

**Published:** 2019-05-16

**Authors:** Paige K. Smith, Melodi G. Sen, Paul R. Fisher, Sarah J. Annesley

**Affiliations:** Department of Physiology, Anatomy and Microbiology, La Trobe University, Bundoora 3086, Melbourne, Australia; paigeksmith81@gmail.com (P.K.S.); mgsen@students.latrobe.edu.au (M.G.S.); P.Fisher@latrobe.edu.au (P.R.F.)

**Keywords:** *Dictyostelium*, CLN2, Tpp1, Batten disease, lysosomal storage disease

## Abstract

The neuronal ceroid lipofuscinoses comprise a group of neurodegenerative disorders with similar clinical manifestations whose precise mechanisms of disease are presently unknown. We created multiple cell lines each with different levels of reduction of expression of the gene coding for the type 2 variant of the disease, Tripeptidyl peptidase (Tpp1), in the cellular slime mould *Dictyostelium discoideum.* Knocking down Tpp1 in *Dictyostelium* resulted in the accumulation of autofluorescent material, a characteristic trait of Batten disease. Phenotypic characterisation of the mutants revealed phenotypic deficiencies in growth and development, whilst endocytic uptake of nutrients was enhanced. Furthermore, the severity of the phenotypes correlated with the expression levels of Tpp1. We propose that the phenotypic defects are due to altered Target of Rapamycin (TOR) signalling. We show that treatment of wild type *Dictyostelium* cells with rapamycin (a specific TOR complex inhibitor) or antisense inhibition of expression of Rheb (Ras homologue enriched in the brain) (an upstream TOR complex activator) phenocopied the Tpp1 mutants. We also show that overexpression of Rheb rescued the defects caused by antisense inhibition of Tpp1. These results suggest that the TOR signalling pathway is responsible for the cytopathological outcomes in the *Dictyostelium* Tpp1 model of Batten disease.

## 1. Introduction

The neuronal ceroid lipofuscinoses (NCLs), also known as Batten disease, are a group of progressive neurodegenerative disorders typically appearing in childhood. Clinically, the disease presents with motor and mental deterioration, visual loss, seizures, ataxia and reduced life span [1]. NCLs are classified as lysosomal storage disorders and are characterised by the accumulation of autofluorescent material in intracellular vesicles. In many forms of NCL, the primary component of this autofluorescent material is mitochondrial ATPase subunit C [2]. Mutations in thirteen different genes have been identified (CLN1–-8 and CLN10–14), and the protein products of most of these are localised to the lysosome, although some are localised to the ER or vesicular membranes [1].

The gene for the type 2 variant, or CLN2, has been identified and found to encode a lysosomal enzyme known as tripeptidyl peptidase 1 (TPP1). Most of the TPP1 mutations identified to date are responsible for the classic late infantile form of the disease [3]. All mutations result in a marked deficiency or loss of TPP1 activity and absence of the functional protein [4]. Onset occurs between two and four years of age and is characterised by an accumulation of fluorescent lipopigment in brain and spinal neurons, skin, muscle and the eyes [3]. Eventually, cell death occurs in these tissues, presenting with deteriorating muscle function, progressive mental deterioration and seizures.

TPP1 is a serine protease responsible for the cleavage of tripeptides from the amino acid terminus of small polypeptides undergoing degradation in the lysosomes [5]. Little is known about the in vivo substrates of TPP1, but, in humans, loss of TPP1 activity leads to neuronal accumulation of subunit c of ATP synthase [6] and accounts for 85% of protein content of the storage material [2]. An increase of TPP1 protein has been reported in other pathological conditions including other neurodegenerative lysosomal storage disorders such as CLN3 disease, as well as inflammation, cancer and aging [7]. Although it is clear that the regulation of this protein is important, little is known about the normal physiological roles of TPP1. 

Model systems provide an invaluable tool for investigating complex disorders such as NCL. They permit the study of disease mechanisms and the development and testing of therapeutic strategies and several reviews have been written about the contribution such models have made to our understanding of NCL [8,9,10,11]. *Dictyostelium discoideum* is one of these models. *Dictyostelium* is a cellular slime mould or social amoeba, it has all the benefits of a model system with a haploid completely sequenced genome, it is genetically tractable and it is amenable to a range of biochemical, cell biological and physiological studies [12]. In addition, *Dictyostelium* has a unique life cycle with unicellular and multicellular stages. It begins the life cycle as a unicellular amoeba feeding on microorganisms in the soil. When the food source is depleted, the cells begin to emit and respond to a chemical signal, cAMP (cyclic adenosine monophosphate). This leads to the formation of an aggregate consisting of approximately 10^5^ cells which undergoes multicellular development with numerous stages leading to the final structure of a fruiting body consisting of a basal disc, a long slender stalk containing cells that have undergone autophagy and a sorus containing spores. This unique life cycle provides a plethora of phenotypes for study which are essentially readouts of the underlying signalling pathways [12].

*Dictyostelium* contains homologues for 11 of the 13 NCL genes. Four CLN gene homologues, CLN10, CLN2, CLN3 and CLN5, have been studied in some detail in *Dictyostelium*. CLN10 encodes Cathepsin D and these null mutants do not display any significant defects in growth or development [13]. CLN5 localises to the cell cortex and is secreted during early multicellular development. This secretion is via an unconventional pathway involving the contractile vacuole (CV) system, is reliant on its glycosylation, and is modulated by CLN3 [14]. A CLN5 knockout strain showed defects in cell–cell and cell–substrate adhesion and a reduction in cAMP-mediated chemotaxis with no effect on aggregation. In addition, CLN5 in *Dictyostelium* was found to have glycoside hydrolase activity similar to its mammalian counterpart [15].

CLN3, a transmembrane protein, was shown to negatively regulate proliferation and development in *Dictyostelium* [16]. Huber et al. [16] created a CLN3 knockout and were able to rescue the phenotypic defects through expression of either *Dictyostelium* or human CLN3, showing a conservation of CLN3 function. CLN3 localises to the contractile vacuole (CV) and the endocytic pathway [17]. The null mutant displays impaired streaming morphology and aggregation [17]. 

CLN2 or TPP1 is well conserved amongst vertebrates, but is lacking in many simple eukaryotes such as *S. cerevisiae, C. elegans* and *Drosophila*. The genome of the simple eukaryote, *Dictyostelium discoideum,* however, does contain TPP1 homologues. In fact, at least six homologues have been identified named *tpp1A* to *tpp1F* [18]. They can be separated into two major groups, the first being the Tpp1A group which includes members Tpp1A–C. These proteins show similar structure to the mammalian homologue. The second group includes Tpp1D-F and all contain a conserved insertion in the peptidase domain [18]. Of the six different CLN2 homologues in *Dictyostelium* two have been studied to date. Tpp1 encoded by the *tpp1A* gene (DDB_GO269914) was studied by Philips and Gomer and a GFP-tagged Tpp1 protein was shown to localise to the lysosome [19]. They created a *tpp1A* gene knockout in which most of the ORF was deleted. The mutant displayed reduced cleavage of a Tpp1 substrate, precocious multicellular development and a reduced ability to form spores. Expression of *Dictyostelium* Tpp1A or human CLN2 rescued the reduction in Tpp1 activity. The authors also demonstrated a role for Tpp1 in inhibiting autophagy, as *tpp1A* null cells showed a reduction in cell size and viability in autophagy stimulating conditions [19]. 

The second *Dictyostelium* CLN2 homologue studied is encoded by the *tpp1F* gene (DDB_GO281823) formerly named *v4-7* after the vegetative stage specific mRNA4-7. Tpp1F is a soluble luminal protein that is localised to the ER and to endolysosomal compartments [18]. Tpp1F mutants do not contain any N-glycosylation sites, in contrast to Tpp1A and mammalian CLN2, and do not display any obvious defects in growth and development [18]. Tpp1F did exhibit Tpp1 activity but no abrogation of Tpp1 activity was observed in the null mutant presumably due to compensation by the other Tpp1 homologues. Tpp1F was identified as an interaction partner of the Golgi pH regulator (GPHR) and Tpp1B has also been found to bind GPHR [18]. Binding to GPHR interferes with Tpp1F activity and a reduced protein level and Tpp1 activity were observed in a GPHR null strain [18]. In an analysis of the conditioned medium (CM), a higher amount of Tpp1F was observed in the CLN3 null mutant compared to wild type [17]. 

To further investigate the role of Tpp1 we have created *Dictyostelium* mutant strains which have reduced expression of Tpp1A through antisense inhibition. We chose Tpp1A initially as it was the only CLN2 homologue identified at the time, but also as this protein displays protein sequence and protein structure conservation to mammalian CLN2, is *N*-glycosylated, displays Tpp1 activity and a previously constructed null mutant displays developmental defects similar to other lysosomal mutants. We created multiple transformants all with differing levels of *tpp1A* expression, enabling comparison of phenotype severity with *tpp1A* expression. Here we describe roles for Tpp1A (henceforth referred to as Tpp1) in growth, endocytosis and development. 

To investigate the mechanism of these defects, we investigated whether altered TOR signalling might be involved. The TOR protein kinase pathway has long been known to play an essential role in sensing and responding to nutrient stress and intracellular energy state [20]. The protein itself is involved in two complexes, TORC1 (Target of Rapamycin Complex 1) and TORC2 (Target of Rapamycin Complex 2). TORC1 is known to promote growth by targeting the phosphorylation of downstream proteins involved in gene expression, protein translation and autophagy [21]. The complex is conserved across eukaryotes, including the yeast *Saccharomyces*, the amoebozoan *Dictyostelium,* invertebrate metazoa such as *C. elegans* and mammals [20]. The main function associated with TORC2 is regulation of the actin cytoskeleton and this function is well conserved across organisms. TORC2 also has a role in endocytosis. Here, we show that pharmacologically inhibiting TOR signalling through Rapamycin treatment and genetically via antisense inhibition of Rheb mimicked the defective phenotypes observed in the Tpp1 antisense inhibited strains. We then created strains in which Tpp1 expression is stably antisense-inhibited and were able to rescue the phenotypic defects by overexpression of Rheb. This implicates the TOR signalling pathway as a possible mechanism for the downstream cytopathological effects of Tpp1 loss.

## 2. Materials and Methods

### 2.1. Creation of Constructs

The Tpp1 antisense inhibition construct was made by cloning a 1361 bp fragment of the *tpp1A* gene using *Eco*RI and *Bam*HI sites within the gene. This DNA fragment contains no significant homology with any of the other *tpp1* genes. The fragment was cloned into the *Eco*RI and *Bam*HI sites of the *Dictyostelium* expression vector pDNeo2 in the antisense orientation under the control of the Actin-6 promoter. The Rheb antisense inhibition construct was made by amplifying a 365 bp fragment of the Rheb gene from bases 193 to 558 using gene specific primers (RhebASfwd: GCGCGAATTCTCTAGAGATGAATATTCAATTTTACAAAAGC and RhebRev: GCGCGAATTCCTGGAGTTACATTAAAATACAACCTTCTTTTTGTCC) and genomic wild type AX2 DNA as template. The fragment was cloned in the antisense orientation into the *Eco*RI site of pDNeo2. The full length Rheb gene was amplified using the primers (RhebOEFwd: GCGCGAATTCATCGATATGGCACCACAAAAACATAGAAAGATCTG and RhebRev), then cloned into the *Cla*I and *Xho*I sites of the *Dictyostelium* expression vector pA15GFP [22] to replace the GFP gene.

### 2.2. Transformation and Cell Culture

AX2 cells were transformed with 20 μg of the Tpp1 or Rheb antisense inhibition or Rheb overexpression constructs using the calcium phosphate DNA precipitation method [23]. Following selection on *Micrococcus luteus* lawns grown on SM (4.1 mM MgSO_4_.7H_2_O, 16.2 mM KH_2_PO_4_, 5.8 mM K_2_HPO_4_, 1.0% (*w*/*v*) agar (Oxoid, Basingstoke, Hampshire, UK) 1.0% (*w*/*v*) bacteriological peptone (Oxoid, Basingstoke, Hampshire, UK), 0.1% (*w*/*v*) yeast extract (Oxoid, Basingstoke, Hampshire, UK), 1.0% (*w*/*v*) Glucose) agar plates containing 20 μg·mL^−1^ G418 [24], transformants were subcultured and maintained on *E. aerogenes* lawns and axenically in HL-5 on an orbital shaker (150 rpm) at 21 °C.

### 2.3. Calculation of Gene Copy Number

The Tpp1 and Rheb antisense construct copy number was determined using the electrotransformation method [25]. Genomic DNA of wild type AX2, HPF261, Tpp1 and Rheb transformant cell lines was prepared using 1 × 10^7^ cells and DNAzol, according to the manufacturer’s instructions. One tenth of the DNA was electroporated into *E. coli* DH5α cells and plated onto NB plates supplemented with Amp (100 µg·mL^−1^) and grown overnight at 37 °C. The number of colonies were counted, which represented the number of constructs and the data were normalised against the control strain HPF261, which is known to have 35 copies of a control construct, the parental vector into which the antisense fragments were cloned. 

The number of copies of Rheb overexpression constructs was determined by quantitative PCR (Polymerase Chain Reaction) using the primers RhebqPCRFP: GCATTTGTTGAATGCTCTGGTA and RhebqPCRRP: GGAGGTTCAGGACCTGTTG and iQ SYBR Green Supermix according to the manufacturer’s instructions (Bio-Rad, Hercules, CA, USA). Genomic AX2 DNA was used to create a standard curve for estimation of the quantity of gDNA, while standard curves for the purified plasmid construct DNA were used for determining the quantity of the inserted plasmid construct. The single copy gene for filamin was amplified as a loading control using the primers FIL1588F (CCCTCAATGATGAAGCC), FIL1688R (CCATCTAAACCTGGACC). The reaction was performed using an iCycler IQ Multicolor Real-Time PCR Detection System (Bio-Rad, Hercules, CA, USA). The calculations of copy numbers for each construct were based on the quantities of the construct and genomic DNA present in each strain and the sizes (in base pairs) of the construct and whole genome. This yielded raw copy numbers, which were then normalised against the copy number for the control parental strain (since it contains a single copy of the Rheb gene). Normalisation of the raw copy numbers results in a more accurate copy number determination, as it accounts for any systemic experiment-wide errors in raw copy numbers that may arise, for example, from small inaccuracies in the calibration curves. 

### 2.4. Determination of Relative mRNA Expression Levels

Semiquantitative real time RT-PCR (Reverse Transcriptase-Polymerase Chain Reaction) was used to determine relative expression levels of Tpp1A and Rheb mRNA. Real time RT-PCR was performed using the SensiFast SYBR and Fluorescein One-Step Kit and the iQ5 real-time PCR detection system. The expression of filamin was also quantitated and used as a loading control. RNA from the wild type, Tpp1 antisense and Rheb overexpression and antisense transformants was prepared using TriZol (Invitrogen, Carlsbad, CA, USA) according to the manufacturer’s instructions. The RNA was DNase treated and 5 µL was used in a 20 µL reaction mixture containing 1 × SensiFast SYBR and Fluorescein One-Step mix (Bioline, Alexandria, NSW, Australia) and gene specific primers (400 nM) FIL1588F (CCCTCAATGATGAAGCC), FIL1688R (CCATCTAAACCTGGACC), RhebqPCRRP, RhebqPCRFP, 407QPCRF (CGTTTTGGTTTGGAACCATC) and 407QPCRR (GCCATTGCCATGATGTATTG). The PCR cycling used 30 s of denaturation at 95 °C, annealing at 55 °C for 30 s and elongation for 30 s at 72 °C. Expression levels were normalised against the filamin levels to adjust for loading and then measured relative to AX2 control cells [26].

### 2.5. Autofluorescence Experiments

*Dictyostelium* wild type and Tpp1 antisense transformants were grown axenically in HL-5 and 1 × 10^6^ cells were harvested by centrifugation at 2000× *g* for 2 min. The pellet was washed twice in SS (10 mM NaCl, 10 mM KCl, and 2.7 mM CaCl_2_), resuspended in 5 mL of LoFlo HL-5 and incubated for 2 h at 21 °C. The cells were harvested by centrifugation and resuspended in 2 mL of 20 mM sodium phosphate buffer and the fluorescence was measured in a Modulus^®^ fluorometer (Turner Biosystems™) using a UV module (excitation 365 nm, emission 410–460 nm) and a Blue Module (excitation 460 nm, emission 515–570 nm).

### 2.6. Growth Experiments

*Dictyostelium* wild type and transformant cell lines were grown axenically in HL-5 with no antibiotics to exponential phase, subcultured into fresh HL-5 medium (no antibiotics) at an initial cell density of 1 × 10^4^ cells·mL^−1^ and counted using a haemocytometer at 8–12 h intervals over a period of 100 h. The cell densities were then analysed by log-linear regression using the R programming environment computer software to determine the generation time during the exponential phase of growth. 

For growth on bacterial lawns a scraping of *Dictyostelium* cells was taken from a colony growing on a *E. aerogenes* lawn using a sterile toothpick and placed into the centre of a Normal Agar plate (20 g·L^−1^ agar (Oxoid, Basingstoke, Hampshire, UK), 1 g·L^−1^ Bacto peptone (Difco, Detroit, MI, USA), 1.1 g·L^−1^ anhydrous glucose, 1.9972 g·L^−1^ KH_2_PO_4_ and 0.356 g·L^−1^ Na_2_HPO_4_.2H_2_O, pH 6.0) containing a lawn of *E. coli* B2. The diameter of the plaque was measured at 8–12 h intervals for an average of 100 h. The recorded values were analysed by linear regression by using the “R” environment for statistical computing and graphics (http://www.R-project.org) to determine the plaque expansion rate.

### 2.7. Endocytosis Experiments

Bacterial uptake by *Dictyostelium* strains was determined by using as prey an *E. coli* strain expressing a fluorescent protein DsRed [27] as previously described [28]. Macropinocytosis assays were performed using fluorescein isothiocyanate (FITC)-dextran (Sigma-Aldrich, average mol. mass 70 kDa, St. Louis, MO, USA) as previously described [28].

### 2.8. Phototaxis Experiments

Qualitative phototaxis tests were performed as described in Annesley and Fisher [29]. A scraping of amoebae from a colony growing on an *E. aerogenes* lawn was transferred to the centre of a charcoal agar plate (5% activated charcoal, 1.0% agar) using a sterile toothpick. Phototaxis was scored after a 48-h incubation at 21 °C with a lateral light source. Slug trails were transferred to PVC discs, stained with Coomassie Blue and digitised.

### 2.9. Calculation of Cell Numbers in A Slug

Wild type AX2 and Tpp1 transformants were grown axenically to a density of 1 × 10^6^ cells·mL^−1^ and 5 × 10^7^ cells were harvested by centrifugation. The wet pellet was resuspended and plated in a 5 cm × 1 cm line on 2 water agar plates. The plates were placed into a black polyvinyl chloride (PVC) box containing a single 4 mm diameter hole and placed in a lighted room for 16–24 h until slugs were formed. The average number of cells present in slugs was determined by picking individual slugs, dissociating them by repeated pipetting in 50 µL of Sterile Saline (10 mM NaCl, 10 mM KCl, and 2.5 mM CaCl_2_) and counting using a haemocytometer. Thirty slugs for each strain were dissociated and counted and averages were calculated.

### 2.10. Rapamycin Experiments

All rapamycin experiments were performed with the following final concentrations of rapamycin 0, 0.5 nM, 5 nM, 50 nM and 500 nM. Rapamycin was added to the normal agar plates for bacterial growth experiments and to water agar for fruiting body morphology experiments prior to pouring of the plates. It was added to the HL-5 medium at the beginning of the experiment when measuring growth rates in liquid medium. AX2 cells were grown in HL-5 medium containing rapamycin for 24 h prior to endocytosis experiments and before inoculation onto water agar plates for slug cell counts.

## 3. Results

### 3.1. Creation of Transformants with Decreased Levels of Tpp1 Expression

In order to explore the role of Tpp1 in *D. discoideum,* we created stable knock down strains by transforming AX2 wild-type cells with an antisense inhibition construct. The region of the gene targeted for antisense inhibition showed no sequence homology to any other gene. The construct integrates into random sites in the genome, accompanied by rolling circle replication of the plasmid to produce tandemly duplicated copies of the construct within each insertion [30]. Each transformant therefore contains a different number of copies ranging typically from just a few to several hundred [30]. We determined the differing copy numbers of the Tpp1 mutants using the electrotransformation method [30]. Reduced expression of Tpp1 was confirmed by qRT-PCR (quantitative Reverse Transcriptase-Polymerase Chain Reaction) which showed that Tpp1 expression decreased with increasing Tpp1 antisense inhibition construct number (Figure 1). We therefore generated multiple transformants each with a different amount of reduction of Tpp1 mRNA.

### 3.2. Dictyostelium Tpp1 Mutants Display Autofluorescence

Lipofuscin is a fluorescent lipid known to accumulate in the neurons of NCL patients. It is the principal marker used to diagnose the disease and, with an excitation at 365 nm and an emission at 600 nm, it is detected under UV light, but not detectable under blue excitation. We used a fluorometer with both blue and UV excitation filters to detect autofluorescence in the *Dictyostelium* Tpp1 mutants. Figure 2 shows that, relative to the same number of cells of the parental control strain AX2, the Tpp1 antisense transformants displayed increased autofluorescence with excitation in the UV spectrum and no elevated fluorescence with excitation in the blue spectrum. Although the identity of the fluorescent material is not known, this is consistent with an accumulation of lipofuscin as in mammalian cells. The degree of autofluorescence in the UV spectrum correlated with Tpp1 antisense inhibition copy number.

### 3.3. Tpp1 is Required for Normal Development, but Not for Phototaxis

Since previous studies have consistently shown that mutants lacking lysosomal enzymes in *Dictyostelium* display defects in development, we wanted to see if this were also true for the *Dictyostelium* Tpp1-inhibited mutants.

With regard to morphology, we observed that *Dictyostelium* Tpp1 mutants developed into multicellular aggregates quicker than wild type strains (data not shown) as reported previously in a Tpp1 knock out mutant [19] and subsequently formed smaller fruiting bodies (Figure 3A). Because of the smaller fruiting body sizes, we also examined if the mutants formed smaller slugs. This was confirmed quantitatively by counting cell numbers per slug for each transformant, the results revealing a copy number-dependent decrease in slug size (Figure 3B). The loss of Tpp1 activity thus results in precocious starvation-induced development of smaller multicellular slugs and fruiting bodies.

Tpp1 is expressed most strongly during multicellular development with a peak of expression at 16 h corresponding to the multicellular slug stage [31]. The multicellular slug is both phototactic and thermotactic, which enables the slug to migrate to the surface of the soil which is the optimal location for spore dispersal. Given the expression profile of Tpp1 we wanted to explore if the *Dictyostelium* Tpp1 antisense inhibited mutants displayed a defect in slug phototaxis. We found no change in the accuracy of phototaxis in the Tpp1 mutants, regardless of copy number (Figure 3C). Although phototactic orientation was unaltered in Tpp1 antisense transformants, the trails left by the slugs were shorter than those of the wild type. This could be due to a combination of slower migration and earlier decisions to cease migration and culminate. Both of these features of slug migration are characteristic of smaller slugs, the earlier culmination arising because the higher surface to volume ratios of smaller slugs result in more rapid loss of NH_3_ from their cells [32,33].

### 3.4. Tpp1 Knockdown Causes Slow Growth but Elevated Rates of Endocytosis

To determine if Tpp1 plays a role in growth and endocytosis, the effect on growth rates of knocking down Tpp1 was observed both on bacterial lawns and in axenic medium. Figure 4A shows that a reduction in Tpp1 expression correlates with slower plaque expansion rates on bacterial lawns. Similarly, reduced Tpp1 expression results in slower growth when mutants are grown in axenic medium (Figure 4B).

Since decreased levels of Tpp1 expression resulted in slower plaque expansion rates on bacterial lawns and longer generation times in liquid medium, we wanted to determine if these phenotypes could be explained by changes in the ability of cells to ingest nutrients by endocytosis. We conducted both phagocytosis and macropinocytosis experiments to measure uptake of both solid and liquid nutrient sources.

The Tpp1 mutants exhibited a copy number-dependent increase in the rates of phagocytosis (Figure 5A) and macropinocytosis (Figure 5B). The increased rates of phagocytosis and macropinocytosis caused by Tpp1 knockdown suggest the presence of compensatory feedback pathways that upregulate endocytosis rates in response to nutrient deprivation.

### 3.5. Pharmacological or Genetic Inhibition of TOR Signalling Phenocopies Loss of Tpp1

The main pathway controlling growth in all eukaryotic organisms is the TOR signalling pathway. The TOR protein kinase pathway has long been known to play an essential role in sensing and responding to nutrient stress and intracellular energy state [20]. The TOR pathway is activated in response to a variety of molecules including amino acids. Since Tpp1 is required for proteolysis in the lysosome, mutants lacking Tpp1 are expected to be impaired in supplying amino acids to the cell from endolysosomal digestion of nutrients. This would inhibit the TOR signalling pathway. TOR is present in two complexes: TORC1 which upregulates cell growth and TORC2 which controls actin polymerisation and regulates endocytosis. If the inhibition of TOR signalling is responsible for the phenotypes observed in Tpp1 mutants, then inhibition of TOR signalling by other means should phenocopy the loss of Tpp1. We therefore investigated if pharmacological or genetic inhibition of TOR signalling would mimic the phenotypic outcomes of Tpp1 knockdown.

It is possible to inhibit TOR signalling pharmacologically through treatment with rapamycin. TORC1 is sensitive to rapamycin treatment and TORC2 is insensitive to acute rapamycin treatment, but is sensitive to prolonged (24 h) rapamycin treatment [34]. Wild type *Dictyostelium* cells exposed to prolonged treatment with rapamycin displayed concentration-dependent decreases in the size of fruiting bodies (Figure 6A–C) and the number of cells per slug (Figure 6D).

Rapamycin treatment also inhibited growth on bacterial lawns (Figure 7A) and in liquid media (Figure 7B), yet increased the rates of phagocytosis (Figure 7C) and macropinocytosis (Figure 7D). The rapamycin-treated cells thus phenocopied Tpp1 antisense mutants.

To inhibit TOR signalling genetically, rather than pharmacologically, we antisense-inhibited expression of Rheb, the upstream activator of TOR. We also overexpressed Rheb to determine what effect this may have on the cells. The copy numbers of the antisense inhibition and overexpression construct were determined in the transformants and corresponding alterations in the expression of Rheb in these strains was confirmed using qRT-PCR (Figure 8).

The Rheb knock down and overexpression transformants were tested for their growth and endocytosis ability. Inhibition of Rheb resulted in slower growth on bacterial lawns (Figure 9A) and in liquid media (Figure 9B) whilst overexpression resulted in a slight increase in growth in both substrates. Rheb antisense inhibition resulted in increased phagocytosis and macropinocytosis rates and conversely overexpression of Rheb resulted in decreased phagocytosis and macropinocytosis rates (Figure 9C,D, respectively). Therefore, a reduction in Rheb expression phenocopied the antisense inhibited Tpp1 transformants.

### 3.6. Overexpression of Rheb Rescues the Phenotypic Effects in the Tpp1 Mutants

To test the hypothesis that Tpp1 can regulate TOR signalling, cotransformants were created which had reduced Tpp1 expression through antisense inhibition and increased Rheb expression through overexpression. The copy numbers of each construct were determined by qPCR. The number of Tpp1 antisense inhibition copy numbers ranged up to 105 and Rheb overexpression copy numbers up to 99. Defective phenotypes were observed in the Tpp1 knockdown strains with construct copy numbers in this range, thus, if no rescue occurred, then defective phenotypes should still be observed in the cotransformants.

Cotransformants produced fruiting bodies of a similar size to wild type and the number of cells per slug were within the wild type range (Figure 10). This indicates that the reduced slug size and smaller fruiting bodies seen in the Tpp1 mutants was rescued by overexpression of Rheb.

Growth on bacterial lawns and in axenic media were measured and the cotransformants displayed growth rates in the wild type range indicating a rescue of the decreased growth rates observed in the Tpp1 mutants by overexpression of Rheb (Figure 11A,B). Likewise, the increased phagocytosis rates observed in the Tpp1 mutants were rescued in the cotransformants which displayed rates in the wild type range (Figure 11C). The macropinocytosis rates of the cotransformants were also not significantly different from wild type and also showed no effect of copy numbers of the Tpp1 antisense construct (p = 0.49), unlike the Tpp1 antisense inhibition on its own (p = 0.047, Figure 11D). This would be consistent with Rheb overexpression also rescuing the effect on macropinocytosis of the Tpp1 knockdown. However, this conclusion remains uncertain because of the relatively small range of Tpp1 construct copy numbers we were able to obtain in the cotransformants, combined with the relatively small effect on macropinocytosis that Tpp1 knockdown has within this copy number range. It would be valuable in future to re-examine this by overexpressing Rheb in a TppA null mutant background.

Overall, these results suggest a role for TOR signalling in the *Dictyostelium* Tpp1 antisense inhibited strains.

## 4. Discussion

In this study, we investigated the biological role of the Batten disease-associated protein Tpp1 in *Dictyostelium* by knocking down expression levels using antisense inhibition. Tpp1 knockdown mutants exhibited increased autofluorescence, the cellular hallmark of Batten’s disease. The autofluorescence observed here is in contrast to Philips and Gomer [19] who could detect autofluorescence in 16 h starved cells but not in vegetative Tpp1 knockout cells. The authors suggested that autofluorescence may not have been observed in the Tpp1 knockout vegetative cells as no mRNA transcripts were detectable in vegetative cells thereby suggesting that any effects of knocking out Tpp1 would only be observed in the developing cells [35]. This has not proved to be the case in other *Dictyostelium* CLN mutants such as CLN3. CLN3 is expressed at low levels in vegetative cells with expression increasing during development and peaking at 12 h post starvation. Despite not being highly expressed in the vegetative stage, a knockout strain showed increased cell proliferation in axenic media [16]. The difference between our autofluorescence results and those of Philips and Gomer [19] could be technical. They used cells grown in the presence of bacteria, whereas we used cells grown in liquid medium. Another, possibly more significant difference was the excitation wavelengths used—in our experiments, we used an excitation wavelength in the UV range (365 nm) in response to which lipofuscin is known to emit a yellow fluorescence. The reported ranges for excitation and emission of lipofuscin in Batten disease cells are 350–380 nm and 400–600 nm, respectively [36,37]. Philips and Gomer [19] used excitation wavelengths of 543 nm for laser confocal experiments and 488 nm for flow cytometry experiments. We could not see any difference in autofluorescent material when using an excitation wavelength of 460 nm. Our results confirm that in *Dictyostelium*, as in human cells, Tpp1 loss causes the accumulation of material that autofluoresces at yellow wavelengths in response to UV light. Autofluorescence has also been observed in *Dictyostelium* CLN5 null mutants [15].

We showed that Tpp1 plays an essential role in growth and multicellular development. The Tpp1 antisense-inhibited transformants produced smaller aggregates and the number of cells in each slug correlated with the amount of inhibition. This is consistent with Philips and Gomer’s [19] Tpp1 (CLN2) knockout and Huber et al. [16] CLN3 knockout models, which also exhibit faster progression through the life cycle, forming smaller fruiting bodies earlier than the wild type. The formation of smaller aggregates and fruiting bodies occurs because more cells reach the point of initiating aggregation, to form aggregation centres that recruit other cells by chemotaxis. This produces more, but smaller aggregates from the starting population of cells. In addition to the small slugs and aggregates, our antisense-inhibited transformants displayed slow growth on bacterial lawns and in axenic media which correlated with the amount of antisense inhibition and suggests that Tpp1 loss causes inefficient lysosomal degradation of nutrient sources. Slower cell growth and altered development is consistent with other Batten disease models including CLN1 and CLN3 knockouts in yeast and premature ageing in a CLN3 knockout worm [38].

It is not surprising that Tpp1 plays a role in suppressing the initiation of normal development so that its loss accelerates development. The developmental transition of *Dictyostelium* from single-celled amoebae to a multicellular organism has been studied in considerable detail and is initiated by nutrient starvation. Given the function of Tpp1 in lysosomal metabolism of cellular waste in mammalian systems, it is expected that this enzyme would play a similar, if not more important, role in *Dictyostelium*, where lysosomes contribute not only to turnover and degradation of defective cellular components, but also to processing of prey particles internalised by endocytosis. Depending on where the enzyme acts in the degradation cascade and whether or not the lysosome contains other enzymes that can replace them functionally, their loss may be well tolerated (e.g., Cathepsin D) [13]. However, the lack or insufficiency of key enzymes would result in inefficient digestion of cellular nutrients, producing partial and/or precocious starvation of affected mutants, even while the external supply of nutrients or bacterial prey remains sufficient for healthy, wild type cells. This was seen in mutants lacking the lysosomal enzyme N-acetyl-hexosaminidase [39]. The corresponding mutant grew slower, and formed small slugs and fruiting bodies, similar to the phenotype we observed here for Tpp1 knockdown mutants. Our results suggest that, given an equal amount of prey, Tpp1 mutants reach starvation earlier due to ineffective nutrient processing, so that multicellular development is initiated earlier by nutritional stress.

The ineffective processing of nutrients may also be the cause of the slower growth rate. The Tpp1 mutants grew slower on bacterial lawns and in axenic medium. Given the Tpp1 mutants’ poor growth, we were initially surprised by the results of the endocytosis assays. The process of endocytic uptake of nutrients by *Dictyostelium* amoebae and subsequent digestion of particles involves multiple events. Particles are bound by a surface receptor followed by changes in the actin cytoskeleton that allow the particle to be surrounded by a cellular membrane. Fusion and internalisation of the particle into endocytic vesicles ensues, followed by lysosomal processing and subsequent degradation of vesicle contents [40].

Although processing and degradation of lysosomal contents in the Tpp1 mutants is expected to be inefficient or impaired, it is not unprecedented that these mutants take up nutrients via endocytosis at faster rates. A *Dictyostelium* mutant lacking the gene *alyA*, that contributes significantly to total lysozyme activity, grows slowly to produce small plaques on a bacterial lawn but, in a compensatory pathway, displays increased rates of phagocytosis of fluorescently labelled yeast cells [41].

Another example of opposing effects on endocytosis and growth is shown in serum response factor B-deficient (*srfB*^-^) cells. These cells show a reduction in several lysosomal proteins including the CLN10 homologue cathepsin D (*ctsD*). The cathepsin D protein is reduced 10-fold in this *srfB*^-^ mutant. These cells display decreased growth rates on bacteria and in axenic media yet display increased macropinocytosis rates [42]. The mutants also form aggregates earlier [42] reminiscent of the Tpp1A and CLN3 *Dictyostelium* mutants. Such compensatory pathways allow healthy cells to upregulate endocytosis when the nutrient supply becomes insufficient to meet demand. When the underlying cause of the nutrient undersupply is genetic and thus permanent, as in these mutants, the compensatory pathways bring the cells to a new steady state in which the demand and supply for nutrients are balanced between decreased growth rates and elevated endocytosis rates. What might be the regulatory pathways that are responsible for this altered steady state in mutant cells?

One mechanism for the decreased growth and increased endocytosis could be through the TOR (Target of Rapamycin) signalling pathways. The TOR protein itself is involved in two complexes, TORC1 and TORC2. In addition to TOR, the two complexes share common, namely Lst8, and unique subunits, such as Raptor in TORC1 as well as Rictor and SIN1 in TORC2. TORC1 is known to promote growth by targeting the phosphorylation of downstream proteins involved in gene expression, protein translation and autophagy [21]. TORC1 is activated in response to a variety of molecules including amino acids, which in turn allows growth promotion within the cell. Corresponding down-regulation of TORC1 in response to nutrient depletion is known to promote diversion into autophagic cell death pathways, but may also stimulate pathways for energy recovery such as proline metabolism to generate ATP [43]. TORC1 is acutely sensitive to rapamycin and is inactivated by it. TORC2 on the other hand is insensitive to acute rapamycin treatment but can be inhibited by prolonged exposure to rapamycin [34].

The main function associated with TORC2 is regulation of the actin cytoskeleton and this function is well conserved across organisms. TORC2 also has a role in endocytosis. In *Dictyostelium* depletion of TORC2 components Pia (Rictor), Sin1 or Lst8, and prolonged exposure to rapamycin, which is known to inhibit TORC2, all led to increased phagocytosis rates, but macropinocytosis was unaffected [44].

To determine if TORC1 and TORC2 signalling could be responsible for the effects of Tpp1 loss, we inhibited the TORC1 and TORC2 complexes by prolonged exposure to rapamycin. The resultant phenotypes mimicked downregulation of Tpp1. Fruiting bodies were more numerous and smaller, while growth in liquid and on bacterial lawns was reduced. These phenotypes mimic what was observed to result from Tpp1 knockdown and are consistent with TORC1′s well established roles in starvation signalling and growth regulation.

Phagocytosis rates were increased by rapamycin treatment as expected from an inhibition of TORC2 [44]. However, macropinocytosis rates were also increased, in contrast with data from Rosel et al. [44] who could not detect an effect on macropinocytosis of exposure to rapamycin. This may be due to differences in the time of exposure to the drug. Rosel et al. [44] exposed their control AX3 strain to 500 nM rapamycin for 5 h, whereas we exposed our control AX2 strain to differing concentrations of Rapamycin (0–500 nM) for 24 h. The longer exposure time was chosen based on experiments by Sarbassov et al. [34] who showed that prolonged rapamycin treatments of 24, 48 and 72 h caused a reduction in TORC2 activity in HeLa and PC3 cell lines. We could also show that this affect was concentration-dependent with the effect on macropinocytosis being more severe at higher concentrations of rapamycin. Rosel et al. [44] did see differences in the effect of rapamycin in multiple phenotypes when cells were exposed for 30 min compared to 5 h, but our results suggest that even longer exposure to rapamycin, 24 h, does effect macropinocytosis. These longer exposure times may better represent the situation in our mutants, in which the TORC2 inhibition resulting from nutrient undersupply results from a permanent genetic defect.

To further investigate TOR signalling, we antisense-inhibited and overexpressed Rheb (Ras homologue enriched in brain). Rheb, a small GTPase is highly conserved amongst eukaryotic organisms [45]. It acts as a positive regulator of TOR signalling, activating TORC1, but little is known about its regulation of TORC2 in any system. There are reports that TORC2 is regulated both by amino acids and Tsc2, the upstream inhibitory GTPase Activating Protein (GAP) for Rheb [46]. Studies in *Dictyostelium* have shown that Rheb negatively regulates phagocytosis in a TORC2-dependent manner [44], while overexpression of Rheb reversed the inactivation of TORC1 by nutrient deprivation. In agreement with this, we found that knocking down Rheb expression resulted in increased phagocytosis rates, consistent with inactivation of TORC2. Rheb knockdown also decreased growth rates on bacterial lawns, consistent with inactivation of TORC1. Overexpression of Rheb resulted in the opposing phenotypes with decreased phagocytosis and increased growth rates.

Antisense inhibition of Rheb also resulted in increased macropinocytosis rates, in contrast with the results of Rosel et al. [44] who observed a slight, but insignificant increase in macropinocytosis in a Rheb null strain. The difference could be a result of different parental strains being used. Rosel et al. [44] used AX3 and we used AX2. These two strains were isolated independently from the original parental isolate NC-4, AX3 after exposure to a mutagen and AX2 after prolonged culture with no added mutagen [47,48]. The two strains display different rates of macropinocytosis and may have differences in regulation of this phenotype.

Similar to rapamycin inhibition of TOR signalling, knockdown of Rheb resulted in phenotypes that mimic loss of Tpp1. To determine if Tpp1 loss activated nutrient stress signalling through TORC1 and TORC2 we created cotransformants which antisense inhibited Tpp1 and also overexpressed Rheb. Overexpression of Rheb rescued the slug size, growth and endocytic phenotypes caused by antisense inhibition of Tpp1. These results demonstrate a genetic interaction between the TOR signalling regulator Rheb and the CLN2 homologue Tpp1. They suggest that Tpp1 is required for the efficient degradation of lysosomal polypeptides and the generation of free amino acids that are important activators of Rheb which in turn activates TORC1 and TORC2. As TORC1 and TORC2 activity were not measured directly, it is possible that Tpp1 knockdown is acting through a shared target protein downstream of the known Rheb, TORC1 and TORC2 pathway but no such target protein is currently known. Tpp1 is involved in lysosomal protein degradation and the amino acid supply (from endolysosomes) is known to regulate TOR signalling. The simplest hypothesis then is that Tpp1 loss is upstream of Rheb and TOR signalling.

TOR signalling has long been suggested to play a role in Batten disease. Our results agree with studies in yeast investigating the CLN3 homologue Btn [49]. Synthetic genetic arrays identified interactions between Btn and the core components of TORC1 and TORC2. The Btn null cells exhibited phenotypes consistent with defective TOR signalling and the Btn-defective phenotypes were rescued by inhibiting the upstream regulator of TORC1, Rhb1 [49]. Our results similarly support a role of TOR signalling in Batten disease, in this case the form associated with loss of CLN2 function.

## Figures and Tables

**Figure 1 cells-08-00469-f001:**
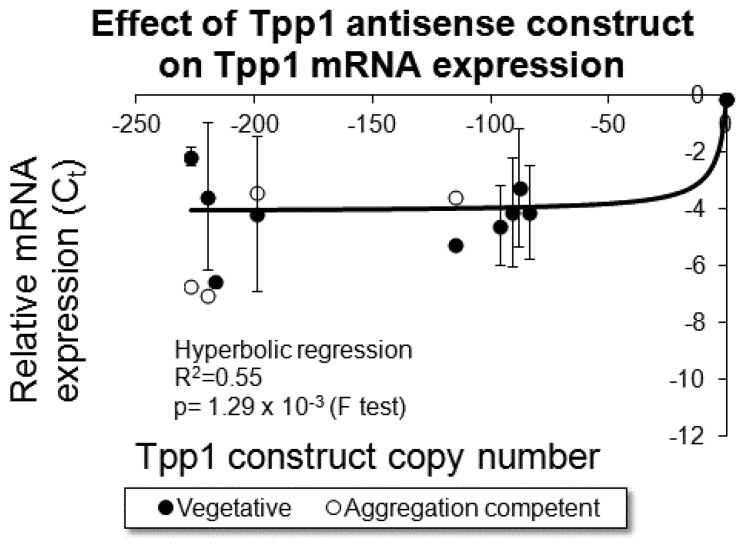
Tpp1 expression was reduced in the transformants. RNA was extracted from cells grown axenically (Vegetative) and cells starved for 8 h (Aggregation competent). Semiquantitative RT-PCR was performed for each of the transformants and the wild type strain and the threshold values obtained were normalised by subtraction relative to the filamin mRNA, and plotted against the measured copy number data. Prior to normalisation the Ct value for amplification of the *tpp1* and filamin gene using the wild type AX2 strain was 21. These data show that, as the number of Tpp1 antisense inhibition constructs increased, the level of *tpp1* mRNA decreased. Negative values were assigned to the Tpp1 antisense inhibition construct to indicate reduced expression of Tpp1. The data are from three independent experiments and error bars represent standard deviation. The regression was significant at indicated p value (F test). The line was fitted by the least squares method to a hyperbolic equation.

**Figure 2 cells-08-00469-f002:**
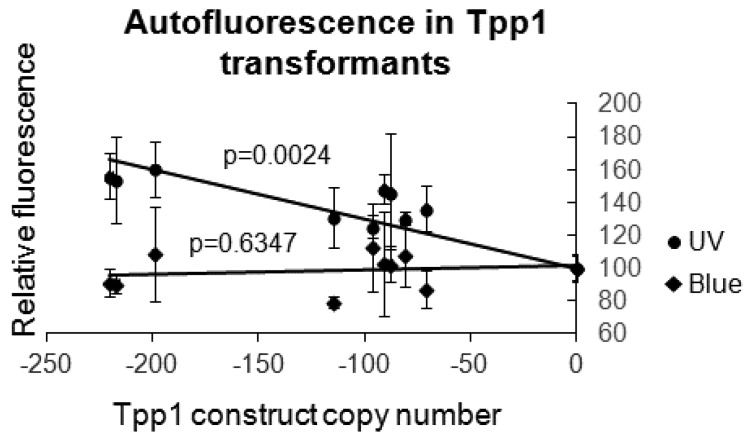
*Dictyostelium* Tpp1 mutants display increased autofluorescence in UV but not blue light. Fluorescence values were measured for wild type AX2 and Tpp1 mutant cells grown in Lo-Flo HL-5 using the UV (excitation λ = 365 nm, emission λ = 410–460 nm) or Blue (excitation λ = 460 nm, emission λ = 515–570 nm) Modules in a Turner modular flourometer. The values were normalised to AX2 and plotted against the Tpp1 antisense inhibition construct copy number. The amount of fluorescence measured with the UV module increased with increasing copy number and was unchanged when measured with the Blue module, as expected for lipofuscin-associated autofluorescence. Data are the mean from three independent experiments and error bars represent standard deviation. The regressions were significant at indicated p values (t test). The lines were fitted by the linear least squares method.

**Figure 3 cells-08-00469-f003:**
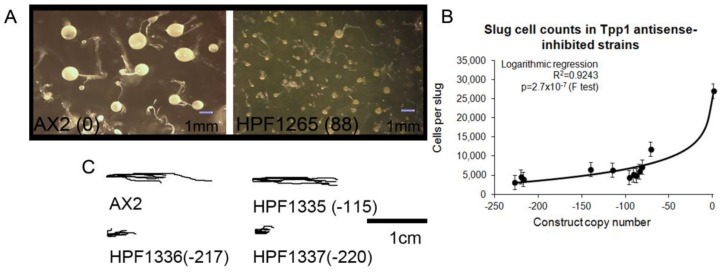
Tpp1 is required for normal development. (**A**) Tpp1 mutant (HPF1265) produced more numerous, smaller fruiting bodies than the wild type (AX2). Numbers in brackets (88) represent the number of Tpp1 antisense inhibition construct. (**B**) The average number of cells per slug for each strain was plotted against the Tpp1 antisense inhibition construct copy number. At least 30 individual slugs for each strain were counted and the means were plotted with error bars representing standard deviation. The regression was significant at indicated p value (t test). The line was fitted by the least squares method to an exponential equation. (**C**) Slug trails of wild type, AX2 and three Tpp1 mutants (HPF1335–1337). The light source is to the right of the figure. Tpp1 mutants produce shorter trails.

**Figure 4 cells-08-00469-f004:**
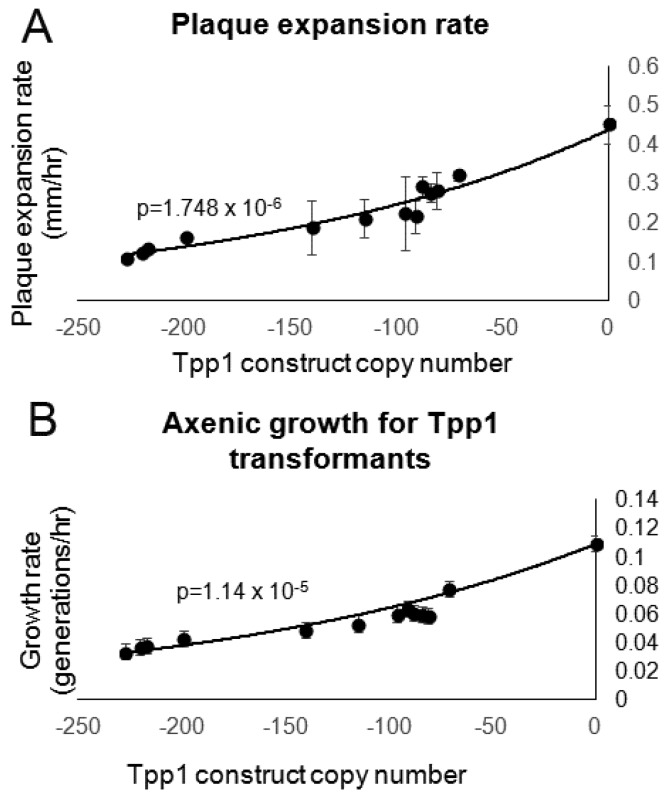
Reduction of Tpp1 inhibits growth. (**A**) Following growth for 4–5 days on *E. aerogenes* lawns, a scraping of *Dictyostelium* cells were removed from the edge of the plaque and transferred to an overnight lawn of *E. coli* B2. The diameter of each *Dictyostelium* plaque was then measured twice daily for five days and the rate of plaque expansion calculated. Experiments were performed in triplicate in three individual experiments. Error bars represent standard deviation. (**B**) Cells were suspended to a final concentration of 1 × 10^4^ cells·mL^−1^ and grown on a shaker at 21 °C. A small amount of medium containing *Dictyostelium* cells was removed and counted twice daily. The growth rate (generations/h) for each strain was then calculated. Experiments were performed twice and error bars represent standard deviations. The regression was significant at the indicated p values (t test). Both lines were fitted by the least squares method to an exponential equation.

**Figure 5 cells-08-00469-f005:**
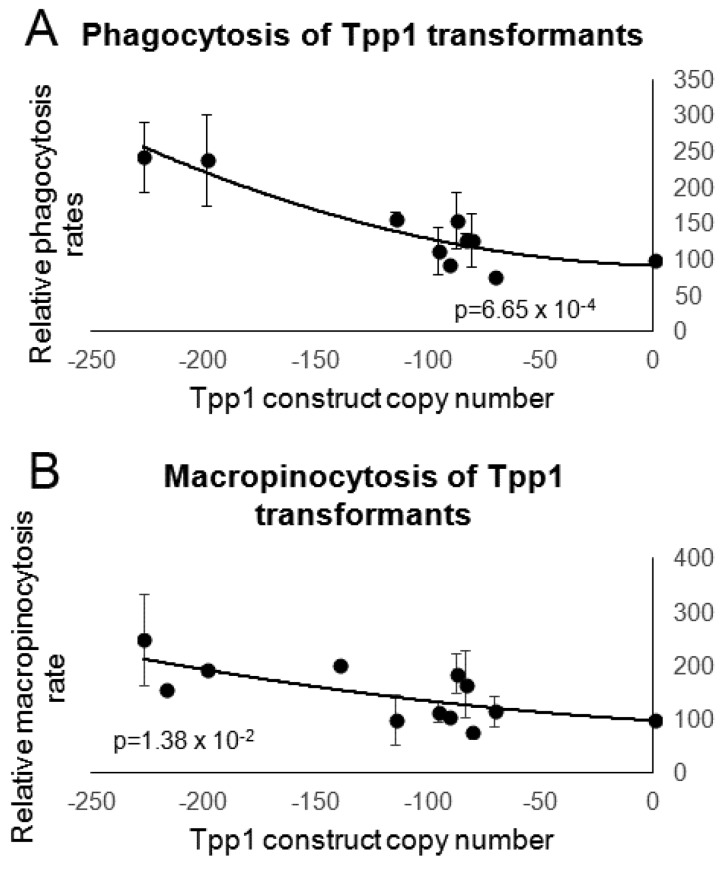
Reduction of Tpp1 increases endocytosis. (**A**) The ability of wild type and Tpp1 mutant cells to ingest a fluorescently labelled bacterium (DsRed-expressing *E. coli*) was measured and results were normalised to wild type. The data show that reduced levels of Tpp1 expression result in increased rates of bacterial uptake. Cell lines were measured in at least three independent experiments and error bars represent standard deviations. (**B**) Wild type and Tpp1 mutant cells were resuspended to a density of 1 × 10^7^ cells·mL^−1^ and allowed to take up HL-5 medium containing the fluorescent marker FITC-dextran. Aliquots were removed at specific time points, washed and lysed to release cell contents. Fluorescence was then measured using a fluorometer and values were normalised against the wild type. Cell lines were measured in at least three independent experiments and error bars represent standard deviations. Regressions were significant at indicated p values (t test). Both lines were fitted by the least squares method to an exponential equation.

**Figure 6 cells-08-00469-f006:**
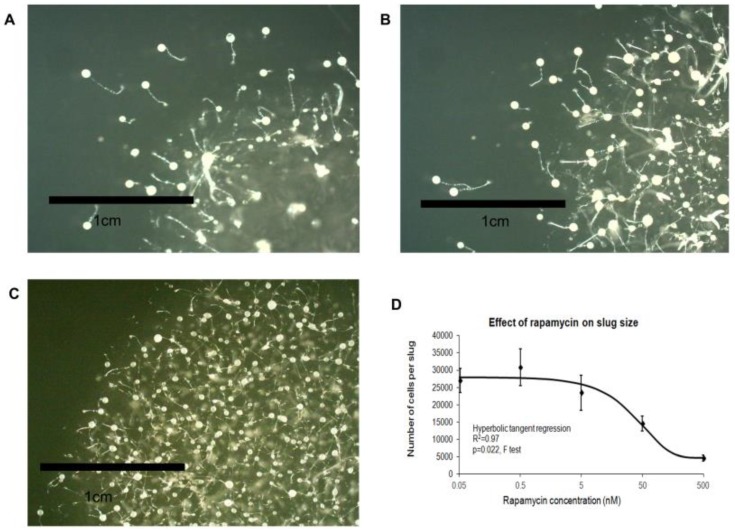
Prolonged rapamycin treatment affects development. Wild type AX2 cells were exposed to rapamycin at: 5 nM (**A**); 50 nM (**B**); and 500 nM (**C**). As the rapamycin concentration increased, the size of the fruiting bodies decreased. (**D**) The average cell counts from 30 slugs for each strain was plotted against the rapamycin concentration with error bars representing standard deviations. The cell counts decreased with increasing rapamycin concentration. The regression was significant at the indicated p value (F test). The line was fitted with the least squares method to a hyperbolic tangent function.

**Figure 7 cells-08-00469-f007:**
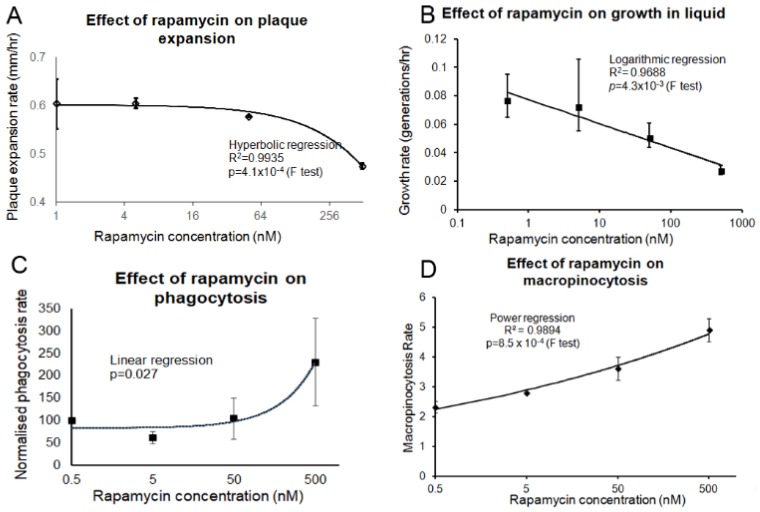
Prolonged rapamycin treatment inhibits growth and promotes endocytosis. (**A**) A scraping of AX2 amoebae grown on an *E. aerogenes* lawn was transferred to NA plates containing rapamycin concentrations ranging from 5 to 500 nM and an overnight lawn of *E. coli* B2. The diameter of each *Dictyostelium* plaque was then measured twice daily for five days and the rate of plaque expansion calculated. The plaque expansion rate was reduced with increasing concentrations of rapamycin. (**B**) AX2 cells were suspended to a final concentration of 1 × 10^4^ cells·mL^−1^ with rapamycin (0.5 nM, 5 nM, 50 nM and 500 nM) and grown on a shaker at 21 °C. A small amount of medium containing *Dictyostelium* cells was removed and counted twice daily. The generation time for each strain was then calculated. The generation time increased with rapamycin concentration. (**C**) Cells in HL-5 medium containing rapamycin (0.5–500 nM) were mixed with fluorescently labelled bacterium (DsRed-expressing *E. coli*) and the rate of uptake was measured and results were normalised to wild type. The rate of phagocytosis was increased with increasing rapamycin concentrations. (**D**) Cells in HL-5 medium containing rapamycin (0.5–500 nM) were allowed to take up HL-5 medium containing the fluorescent marker FITC-dextran. Aliquots were removed at specific time points, washed and lysed to release cell contents. Fluorescence was then measured using a fluorometer. Increasing rapamycin concentrations correlated with increased macropinocytosis rates. The regression for all the graphs was significant at the indicated p values. The lines were fitted with the least squares method.

**Figure 8 cells-08-00469-f008:**
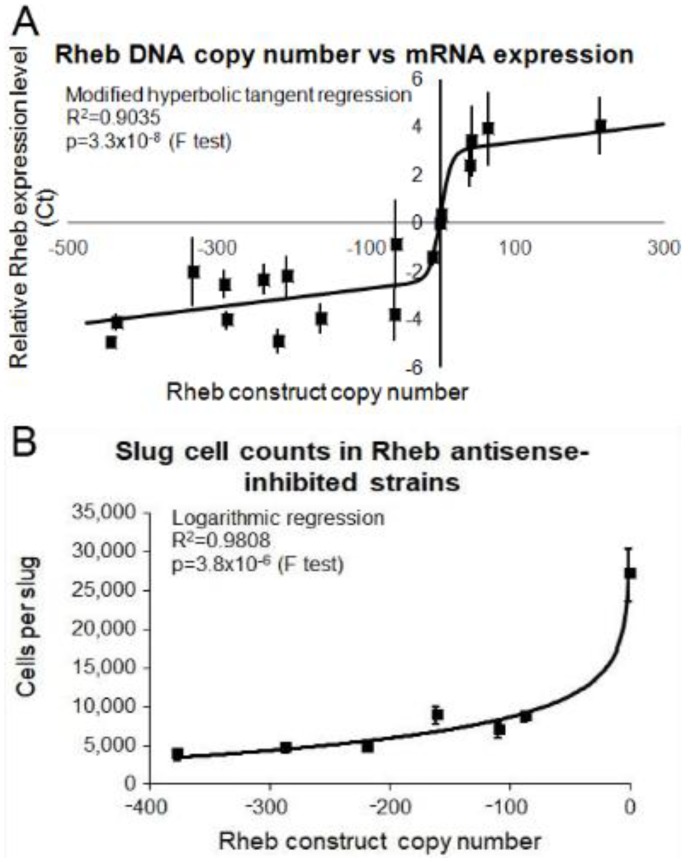
Correlation of Rheb antisense and overexpression copy numbers with Rheb mRNA expression. (**A**) Semiquantitative RT-PCR was performed for each of the wild type, Rheb antisense and Rheb overexpression transformants and the threshold values obtained were plotted against the measured copy number data. The experiments were repeated three times and error bars are generally too small to be seen on the graph. The regression was significant at the indicated p value (F test). The line was fitted by the least squares method to a modified hyperbolic tangent function. These data show that as the number of Rheb antisense inhibition constructs increased the level of Rheb mRNA decreased and vice versa. (**B**) The average number of cells per slug for each strain was plotted against the Rheb antisense inhibition construct copy number. At least 30 individual slugs for each strain were counted and the means were plotted with error bars representing standard deviation. The regression was significant at indicated p value (F test). The line was fitted by the least squares method to an exponential equation.

**Figure 9 cells-08-00469-f009:**
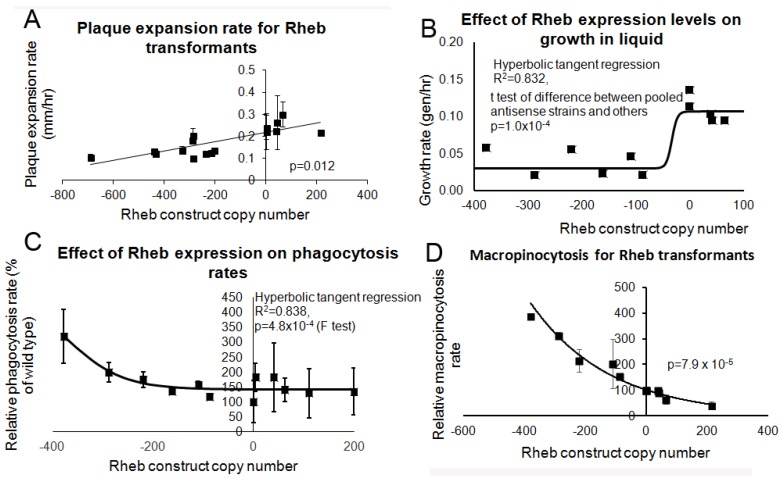
Rheb promotes growth and inhibits endocytosis. Wild type, Rheb antisense and Rheb overexpression transformants were measured for their growth ability and their endocytosis rates. Negative copy numbers represent the number of Rheb antisense constructs and positive copy number values represent the number of Rheb overexpression constructs (**A**) Cells were grown on *E. coli* B2 lawns and the diameter of each *Dictyostelium* plaque was measured twice daily for five days and the rate of plaque expansion calculated. (**B**) Cells were suspended to a final concentration of 1 × 10^4^ cells·mL^−1^ and grown on a shaker at 21 °C. A small amount of medium containing *Dictyostelium* cells was removed and counted twice daily. The growth rate for each strain (generations/h) was then calculated. The growth rates in (A,B) correlated with the number of Rheb construct copy number showing that Rheb positively regulates growth. (**C**) The ability of cells to ingest a fluorescently labelled bacterium (DsRed-expressing *E. coli*) was measured and results were normalised to wild type. The data show that reduced levels of Rheb expression result in increased rates of bacterial uptake. (**D**) The rate of uptake of medium containing the fluorescent marker FITC-dextran was measured and values were normalised against the wild type. The endocytosis rates in (C,D) show that Rheb negatively regulates endocytosis. Each point represents the mean from at least three independent experiments. The regressions were significant at the indicated p values (*t* test in (A,D); and F test in (B,C). Lines were fitted by the least squares method to: a straight line (A); a modified hyperbolic tangent (B,C); and an exponential function (D).

**Figure 10 cells-08-00469-f010:**
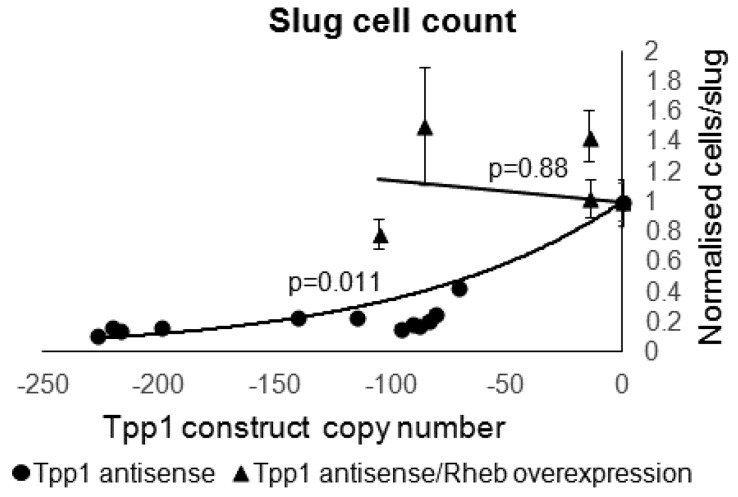
Overexpression of Rheb rescued the decrease in slug sizes caused by Tpp1 knockdown. The average number of cells per slug for each strain was plotted against the Tpp1 antisense inhibition construct copy number. At least 30 slugs were counted for each strain and the error bars represent standard error. The regression was significant at the indicated p values (t test). The line for the Tpp1 antisense strains was fitted by the least squares method to an exponential equation. The line for the Tpp1 antisense/Rheb overexpression transformants was fitted by were fitted by the least squares method to a straight line.

**Figure 11 cells-08-00469-f011:**
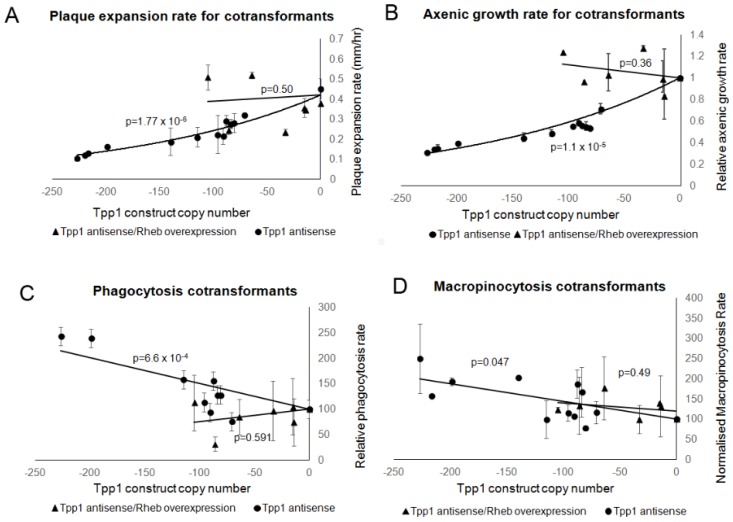
Overexpression of Rheb rescued the growth and endocytic defects caused by antisense inhibition of Tpp1. Wild type, Tpp1 antisense and Tpp1 antisense/Rheb overexpression cotransformants were measured for their growth ability and their endocytosis rates. Negative copy numbers represent the number of Tpp1 antisense constructs (**A**) Cells were grown on *E. coli* B2 lawns and the diameter of each *Dictyostelium* plaque was measured twice daily for 5 days and the rate of plaque expansion calculated. Three replicates were used for each strain and the experiments were repeated three times. Error bars represent standard error. (**B**) Cells were suspended to a final concentration of 1 × 10^4^ cells·mL^−1^ and grown on a shaker at 21 °C. A small amount of medium containing *Dictyostelium* cells was removed and counted twice daily. Error bars represent standard errors. The experiment was repeated twice. The growth rate for each strain (generations/h) was then calculated. The data in (A,B) show that, whilst growth is significantly affected in the Tpp1 antisense transformants, no significant difference in growth was observed in the Tpp1 antisense/Rheb overexpression transformants and wild type cells indicating that overexpression of Rheb rescues the Tpp1 mediated growth defect. (**C**) The ability of cells to ingest a fluorescently labelled bacterium (DsRed-expressing *E. coli*) was measured and results were normalised to wild type. The experiments were repeated a minimum of three times and error bars represent standard error. (**D**) The rate of uptake of medium containing the fluorescent marker FITC-dextran was measured and values were normalised against the wild type. The experiments were performed at least three times and error bars represent standard errors. Data in (C,D) show that endocytosis was significantly affected in the Tpp1 antisense inhibited strains and this defect was rescued by overexpression of Rheb. The regressions were significant at the indicated p values (t test). The lines for the Tpp1 antisense/Rheb overexpression cotransformants in all the panels were fitted by the linear least squares method. The lines for the Tpp1 antisense strains in (A,B,D) were fitted by the least squares method to an exponential equation and in (C) to a linear equation.

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
