# Peer review of "Modelling of Neuronal Ceroid Lipofuscinosis Type 2 in Dictyostelium discoideum Suggests That Cytopathological Outcomes Result from Altered TOR Signalling"

_cells, 2019, doi:10.3390/cells8050469_

Round 1

Reviewer 1 Report

This manuscript uses Dictyosteliumas a model organism to study the role of loss of TPP1 in causing Batten disease in humans. The manuscript describes using antisense methods to cause a reduction in expression of tpp1mRNA in Dictyosteliumcorelating with reduced growth, pinocytosis and phagocytosis, as well as developmental effects. Importantly these phenotypes are also seen on reduction of expression of rheband in the presence of rapamycin, a TOR complex inhibitor. Interestingly overexpressing rheb could rescue the phenotypes of antisense reduction of tpp1expression. This is a novel result potentially linking TPP1 and TOR signalling, suggesting a role or TOR in Batten disease.

Specific points

1.    The autofluorescence detected in growing cells differs from previously published data using a tpp1knock-out strain grown on bacteria. The authors need to demonstrate that this is not due to residual autofluorescence from the growth media in combination with the altered pinocytosis described later in the manuscript. The cells were put into Lo-Flo media for 2 hours prior to measurements but, firstly, is this sufficient to clear any residual HL5, which is known to be fluorescent and secondly, does Lo-Flo media have fluorescence at these wavelengths? 

2.    How long was the axenic growth rate monitored for in figure 4B? Could there be a difference in stationary phase cell number contributing to the phenotype?

3.    The authors interpret the data in Figure 5 as an increase in the rate of phagocytosis and pinocytosis correlating with decreased expression of Tpp1. It is not clear how long the uptake was measured for in either case and how this relates to vesicle flux in wild-type cells – could this reflect a similar rate of uptake but decreased processing downstream, leading to a build-up of material? This could be consistent with lysosomal defects. 

4.    In my download I cannot see alot of the data in Figure 7 as Figure 8A is superimposed on top of it ( I tried a few downloads). However, from what I can see it is not convincing that 5nM (or even really 50nM) rapamycin is having any effect on on phagocytosis and not much on pinocytosis. Is the pinocytosis effect significant at either 5 or 50nM? If these concentrations are not significantly giving any phenotype, they should be removed, but I cannot see the data in panels A and B.

5.    I assume it is for technical reasons that it is not possible to take the strains with a characterised phenotype due to antisense tpp1 expression and overexpress Rheb in these to rescue the phenotypes. Has the reduction of expression of tpp1 mRNA been verified by qRT-PCR in these double construct strains? Otherwise rheb could function by up-regulating tpp1 expression, perhaps by reducing the effectiveness of the antisense.

6.    The authors conclude that reduction in expression of TPP1 leads to changes in TOR signalling. Although that is one interpretation of the data, they cannot definitively conclude this without showing that their antisense tpp1strains show altered phosphorylation of TOR substrates. TORC1 activity can be measured by western blot using phospho-4EBP1 antibody (e.g. Rosel et al.,(2012) J Cell Sci 125, 37). Otherwise TOR signalling could be a parallel pathway. Either this data should be included or the conclusion should reflect alternative explanations. 

7.    The authors use ‘pinocytosis’ and ‘macropinocytosis’ in different places in the manuscript. I believe they are talking about one process so should be consistent or make the distinction clear. 

Author Response

We would like to thank the reviewer for their time and comments and we have addressed each point as below:

This manuscript uses Dictyostelium as a model organism to study the role of loss of TPP1 in causing Batten disease in humans. The manuscript describes using antisense methods to cause a reduction in expression of tpp1mRNA in Dictyostelium corelating with reduced growth, pinocytosis and phagocytosis, as well as developmental effects. Importantly these phenotypes are also seen on reduction of expression of rheb and in the presence of rapamycin, a TOR complex inhibitor. Interestingly overexpressing rheb could rescue the phenotypes of antisense reduction of tpp1expression. This is a novel result potentially linking TPP1 and TOR signalling, suggesting a role or TOR in Batten disease.

Specific points

1.    The autofluorescence detected in growing cells differs from previously published data using a tpp1knock-out strain grown on bacteria. The authors need to demonstrate that this is not due to residual autofluorescence from the growth media in combination with the altered pinocytosis described later in the manuscript. The cells were put into Lo-Flo media for 2 hours prior to measurements but, firstly, is this sufficient to clear any residual HL5, which is known to be fluorescent and secondly, does Lo-Flo media have fluorescence at these wavelengths? 

In Dictyostelium removal of medium via exocytosis has occurred by 2 hours and endosomes containing media have been reported to disappear by 2 hours. We measured the fluorescence of Lo-Flo HL-5 and HL-5 in the fluorometer using the UV and Blue modules. Lo Flo HL-5 displays less than 4% of the fluorescence of normal HL-5 using both wavelengths. If residual medium was present in the cells and accumulating in the knockdown strains, then we would expect the values using both the UV and Blue modules to increase with the degree of antisense inhibition, but this was not the case.

2.    How long was the axenic growth rate monitored for in figure 4B? Could there be a difference in stationary phase cell number contributing to the phenotype?

No. The cells were first grown to exponential phase and then transferred to fresh medium where their growth was measured and recorded for 100 hours as mentioned in the methods section. The growth rate was then calculated only during the exponential phase of growth (the linear portion on a log linear plot). We have not measured their stationary phase and as all strains were first grown to exponential phase any differences in stationary or lag phase were not measured.

3.    The authors interpret the data in Figure 5 as an increase in the rate of phagocytosis and pinocytosis correlating with decreased expression of Tpp1. It is not clear how long the uptake was measured for in either case and how this relates to vesicle flux in wild-type cells – could this reflect a similar rate of uptake but decreased processing downstream, leading to a build-up of material? This could be consistent with lysosomal defects. 

Uptake of E. coli DSRed for phagocytosis was measured over 30 minutes and uptake of FITC-Dextran for pinocytosis was measured over 70 minutes. Both of these times were chosen based on published methods and reflect the time taken for initial uptake from the media. FITC fluorescence bleaches once the endolysosome becomes acidic, so a build-up of material at those stages after the early endosome is not reflected in the signal. In addition, a steady state balance between pinocytosis and exocytosis has been reported to occur only after 90 minutes.

4.    In my download I cannot see a lot of the data in Figure 7 as Figure 8A is superimposed on top of it (I tried a few downloads). However, from what I can see it is not convincing that 5nM (or even really 50nM) rapamycin is having any effect on on phagocytosis and not much on pinocytosis. Is the pinocytosis effect significant at either 5 or 50nM? If these concentrations are not significantly giving any phenotype, they should be removed, but I cannot see the data in panels A and B.

Firstly, we apologise for the superimposition of Figure 8A. The reviewer is correct that at 5 nM Rapamycin the effects were negligible however all concentrations were included to show the dose response relationship, since dose-response curves ideally cover the range from no effect up to maximum effect to produce typically a sigmoidal relationship.

5.    I assume it is for technical reasons that it is not possible to take the strains with a characterised phenotype due to antisense tpp1 expression and overexpress Rheb in these to rescue the phenotypes. Has the reduction of expression of tpp1 mRNA been verified by qRT-PCR in these double construct strains? Otherwise rheb could function by up-regulating tpp1 expression, perhaps by reducing the effectiveness of the antisense.

The reviewer is correct we cannot overexpress Rheb in the existing tpp1 antisense inhibited strains as both constructs are G418 resistant. In addition we could not overexpress Rheb, it appeared to be lethal. Again the reviewer is correct and verifying the levels of tpp1 mRNA would be worthwhile and we could do that given more time to grow the strains from storage extract the RNA and perform the qRTPCR (estimated three weeks).

6.    The authors conclude that reduction in expression of TPP1 leads to changes in TOR signalling. Although that is one interpretation of the data, they cannot definitively conclude this without showing that their antisense tpp1strains show altered phosphorylation of TOR substrates. TORC1 activity can be measured by western blot using phospho-4EBP1 antibody (e.g. Rosel et al.,(2012) J Cell Sci 125, 37). Otherwise TOR signalling could be a parallel pathway. Either this data should be included or the conclusion should reflect alternative explanations. 

We agree with the reviewer and have tried to measure TORC1 activity using the antibodies described in Rosel et al, however in our hands these antibodies did not work. The reviewer is correct that TOR signalling could be a parallel pathway but our data provide no evidence to support this. If TOR signalling was a parallel pathway then Tpp1 and TOR would both need to exert their effects through a shared downstream pathway which is upstream of the multiple phenotypes tested such as growth, endocytosis and slug cell size. We have adjusted the text to reflect alternative explanations as the reviewer suggested as below:

“As TORC1 and TORC2 activity were not measured directly, it is possible that Tpp1 knockdown is acting through a shared target protein downstream of the known Rheb, TORC1 and TORC2 pathway but no such target protein is currently known. Tpp1 is involved in lysosomal protein degradation and the amino acid supply (from endolysosomes) is known to regulate TOR signalling. The simplest hypothesis then is that Tpp1 loss is upstream of Rheb and TOR signalling.”

7.    The authors use ‘pinocytosis’ and ‘macropinocytosis’ in different places in the manuscript. I believe they are talking about one process so should be consistent or make the distinction clear.

We apologise for this inconsistency and have corrected this throughout the document. Pinocytosis has been replaced with micropinocytosis

Reviewer 2 Report

In this study, the authors generate Tpp1A antisense transformants in Dictyostelium and show: enhanced cellular autofluorescence; enhanced endocytic activity (phagocytosis and pinocytosis); slowed vegetative growth; and developmental defects (precocious aggregation and smaller fruiting bodies).  They further show that TOR inhibition (via rapamycin, or rheb antisense transformants) in several ways phenocopies Tpp1A antisense transformants.  Thus the authors suggest that Tpp1A has a role in TOR signaling pathways.

Several pieces of data are strong, showing significant and interesting effects (quite striking effect on fruiting body size).  However, there are open questions that require resolution before this study would be acceptable for publication.

A major concern is poor data correlating Tpp1A transcript abundance with antisense copy number. Only vegetative cells were examined for Tpp1a transcript abundance, and the results as presented are difficult to interpret.  Since it has been reported that TPP1A expression is low/absent during vegetative growth, while it is induced with development, peaking at 16h (Stumpf et al. 2017), it would be logical to assess transcript abundance at developmental stages.  Are RT-PCR primers outside of the 1361 bp antisense sequence, to avoid possible amplification of antisense transcripts?

Additional comments/concerns are listed below.

1) What is the promoter in the pDneo2 vector, and is it active in vegetative and developmental stages? What is the HPF261 line? It has 35 copies of what?

2) Might the 1361 bp antisense have enough sequence similarity with homologs to reduce their expression levels? RT-qPCR should be performed for the other TPP1 homologs in addition to TPP1A, with RNA samples collected from vegetative and developmental stages. Phenotypes at vegetative stages could be due to “off target” knock-down of a homolog.

3) Why do the authors describe their qRT-PCR as semiquantitative, when it is quantitative?  (provides relative or absolute quantitative data).

4) Figure 1 is meant to assess a relationship between antisense copy number and Tpp1A transcript abundance, but the manner of displaying this data seems odd.  Why not simply plot “antisense copy number” (as positive numbers, not negative numbers) on the x axis and relative transcript abundance on the y axis? The use of “expression index” as a label is confusing if one is not aware of “the convention”, particularly when “expression” is on the Y axis.  Also, relative transcript abundance should be plotted relative to WT, where WT = “1” (or 100%), for ease of understanding.

5) Lipofuscin is insoluble aggregates of oxidized proteins, lipids and carbohydrates, and should show broad ex/em fluorescence, under blue as well as UV excitation. Autofluorescence in cells does not necessarily mean lipofuscin.  Increased cellular autofluorescence might merely reflect metabolic shift, with increased levels of naturally fluorescent molecules.  Do the Rheb anti-sense transformants similarly show increased autofluorescence?

Is the autofluorescence intensified after change to developmental (nutrient poor) media? Perhaps the authors could stain for ATP synthase subunit c (Neufeld Ab to detect the aggregated form). As the major protein component of Batten lipofuscin, this would be a better disease correlate.  ATP synthase subunit c aggregates should intensify over time during post-mitotic development, in absence/knock-down of TPP1A.

6) Rather than co-transform, could the authors have performed sequential transformation, for more a direct assessment of phenotypic rescue?

7) The statement “the only experimental evidence for the role of TOR signaling in Batten disease came from studies in yeast…”, is not accurate.  Several CLN3 and CLN2 studies show effects on TOR signaling pathways, in relation to autophagy and/or signaling complex formation at the lysosome (eg. from Cotman, Sardiello labs; and a 2013 paper by Vidal-Donet et al.).

8) the title should specify “CLN2 Batten disease”.

Author Response

We would like to thank the reviewer for their time and comments and have addressed each of the points below:

In this study, the authors generate Tpp1A antisense transformants in Dictyostelium and show: enhanced cellular autofluorescence; enhanced endocytic activity (phagocytosis and pinocytosis); slowed vegetative growth; and developmental defects (precocious aggregation and smaller fruiting bodies).  They further show that TOR inhibition (via rapamycin, or rheb antisense transformants) in several ways phenocopies Tpp1A antisense transformants.  Thus the authors suggest that Tpp1A has a role in TOR signaling pathways.

Several pieces of data are strong, showing significant and interesting effects (quite striking effect on fruiting body size).  However, there are open questions that require resolution before this study would be acceptable for publication.

A major concern is poor data correlating Tpp1A transcript abundance with antisense copy number. Only vegetative cells were examined for Tpp1a transcript abundance, and the results as presented are difficult to interpret.  Since it has been reported that TPP1A expression is low/absent during vegetative growth, while it is induced with development, peaking at 16h (Stumpf et al. 2017), it would be logical to assess transcript abundance at developmental stages.  Are RT-PCR primers outside of the 1361 bp antisense sequence, to avoid possible amplification of antisense transcripts?

The reviewer is correct that the transcript of Tpp1A is reported to be very low in the vegetative stage, yet our results show that despite this low expression there are phenotypic effects of further reducing Tpp1A expression. For this reason, and because many of the phenotypes that we measured were done using vegetative cells, we thought it appropriate (and technically easier) to measure RNA levels in this stage. We were able measure RNA levels at the vegetative stage (thus it is not zero) and we could see a significant 32-64 fold reduction in Tpp1 expression.

Again, the reviewer is correct and RT-PCR primers could, in principle, amplify antisense transcripts. The RT stage of the experiment had only the reverse primer so only the mRNA could be reverse transcribed. The RT itself is not thermostable, so is destroyed in the PCR stage of the experiment. However, low levels of reverse transcriptase activity in the thermostable DNA polymerase in the PCR step can occasionally result in the antisense transcript also being amplified, as the reviewer suggests.  In our experience with many genes and many primer sets, this problem can occur occasionally but is unusual and evidently did not occur here. If it had then we would expect to see increased qRT-PCR signal in the Tpp1 antisense transformants rather than the observed decrease.

Additional comments/concerns are listed below.

1) What is the promoter in the pDneo2 vector, and is it active in vegetative and developmental stages? What is the HPF261 line? It has 35 copies of what?

 The pDNeo2 promoter controlling the expression of the inserted gene is the Actin 6 promoter. A paper by Knecht et al., (1986) Mol. Cell. Biol. shows that expression of proteins from this promoter results in high expression at 0-6 hours of development and this is reduced down to low levels (but not to zero) at 20 hours of development. This indicates that the antisense mRNA would be expressed more strongly during the vegetative and early stages of development. In this case that may help with successful antisense inhibition of the lower concentrations of the target mRNA by increasing the concentrations of the inhibitor mRNA in vegetative cells compared to cells in the multicellular stages. At the later stages the high copy numbers of the construct will also offset any downregulation of the promoter in the construct. The fact that we see phenotypes at both stages indicates that the knock down is effective at both stages.

HPF261 is a control strain which has a known number of integrated constructs (35 copies), it is used to normalise the data. We have added “of a control construct – the parental vector into which the antisense fragments were cloned.” to clarify this.

2) Might the 1361 bp antisense have enough sequence similarity with homologs to reduce their expression levels? RT-qPCR should be performed for the other TPP1 homologs in addition to TPP1A, with RNA samples collected from vegetative and developmental stages. Phenotypes at vegetative stages could be due to “off target” knock-down of a homolog.

 The Tpp1A sequence and the fragment used for antisense inhibition show no sequence homology at the DNA level with any other gene including all of the Tpp1 isoforms. The conservation between the Tpp1 isoforms is at the protein level and not at the DNA level.

3) Why do the authors describe their qRT-PCR as semiquantitative, when it is quantitative? (provides relative or absolute quantitative data).

Because “relative quantitative data” would refer to quantitative data for one gene that has been expressed relative to quantitative data for another gene. We did not use calibration curves to determine quantitative expression levels for the internal control housekeeping gene (filamin) or for the gene of interest. The expression levels are thus measured from differences in Ct cycle numbers between the test and the control gene. These are therefore “semiquantitative” not “relative quantitative”.

4) Figure 1 is meant to assess a relationship between antisense copy number and Tpp1A transcript abundance, but the manner of displaying this data seems odd.  Why not simply plot “antisense copy number” (as positive numbers, not negative numbers) on the x axis and relative transcript abundance on the y axis? The use of “expression index” as a label is confusing if one is not aware of “the convention”, particularly when “expression” is on the Y axis.  Also, relative transcript abundance should be plotted relative to WT, where WT = “1” (or 100%), for ease of understanding.

The number of antisense copy numbers is plotted as a negative number because further in the paper we plot the Rheb antisense construct copy number and Rheb overexpression copy number on the same graph. The negative numbers correspond to antisense inhibition as it is reducing expression and the positive numbers to overexpression as it is increasing Tpp1 levels. For consistency and to follow the convention used in this kind of work established by Bokko et al. (2007), we have kept all antisense construct numbers as negative numbers. This way the expression levels always increase as one moves from left to right on the X axis. In the case of Figure 1 we have changed the X axis title to construct copy number as suggested by the reviewers. The AX2 control value is plotted at 0 because qPCR cycle numbers are a logarithm of the increase in the amount of the template (which grows exponentially). So the correct way to normalize against the control is not by division but by subtraction (the logarithmic equivalent of division). A value of zero means no change and is the logarithm of 1 (which would be the value if the cycle numbers had been converted to template quantities and division used to normalize).

5) Lipofuscin is insoluble aggregates of oxidized proteins, lipids and carbohydrates, and should show broad ex/em fluorescence, under blue as well as UV excitation. Autofluorescence in cells does not necessarily mean lipofuscin.  Increased cellular autofluorescence might merely reflect metabolic shift, with increased levels of naturally fluorescent molecules.  Do the Rheb anti-sense transformants similarly show increased autofluorescence?

The published ranges for excitation emission spectra for lipofuscin in Batten disease cells are 350-380nm and 400-600 nm respectively (Seehafer and Pearce 2006 [45]). We did not observe a difference in the Blue wavelength in agreement with Huber and Mathavarajah, 2018, who also observed autofluorescence in the CLN5 null mutant with excitation: 340–380 nm, emission: 440–480 nm but not with excitation: 475–495 nm, emission: 565–615 nm.

We agree with the reviewers that autofluorescence in these cells does not necessarily mean lipofuscin and have changed the sentence to clarify this as shown below:

“Although the identity of this fluorescent material is not known, this is consistent with an accumulation of lipofuscin as in mammalian cells”.

We have not measured autofluorescence in the Rheb antisense transformants.

Is the autofluorescence intensified after change to developmental (nutrient poor) media? Perhaps the authors could stain for ATP synthase subunit c (Neufeld Ab to detect the aggregated form). As the major protein component of Batten lipofuscin, this would be a better disease correlate.  ATP synthase subunit c aggregates should intensify over time during post-mitotic development, in absence/knock-down of TPP1A.

The antibody described by the reviewer is developed against the mouse protein. The sequence used for development of this antibody when used in a BLAST search against the Dictyostelium protein database does not detect any similar proteins.

6) Rather than co-transform, could the authors have performed sequential transformation, for more a direct assessment of phenotypic rescue?

Unfortunately this is not possible using the constructs we have, as both contain the same selectable marker.

7) The statement “the only experimental evidence for the role of TOR signaling in Batten disease came from studies in yeast…”, is not accurate.  Several CLN3 and CLN2 studies show effects on TOR signaling pathways, in relation to autophagy and/or signaling complex formation at the lysosome (eg. from Cotman, Sardiello labs; and a 2013 paper by Vidal-Donet et al.).

 We apologise for this oversight and have removed the sentences which indicated this. The final paragraph of the discussion has been changed as below:

“TOR signalling has previously been suggested to play a role in Batten disease. Our results agree with studies in yeast investigating the CLN3 homolog Btn [56]. Synthetic genetic arrays identified interactions between Btn and the core components of TORC1 and TORC2. The Btn null cells exhibited phenotypes consistent with defective TOR signalling [56]. Our results similarly support a role of TOR signalling in Batten disease, in this case the form associated with loss of CLN2 function.”

8) the title should specify “CLN2 Batten disease”.

We have changed the title as below:

“Modelling of neuronal ceroid lipofuscinosis type 2 in Dictyostelium discoideum suggests that cytopathological outcomes result from altered TOR signalling”

Reviewer 3 Report

This study provides insight to the function of Tpp1 in Dictyostelium. Antisense inhibition was used to knockdown tpp1 expression, which affected growth, endo/phagocytosis, and development. The authors then link the knockdown phenotypes to TOR signalling.

The title of the paper should be revised since the findings suggest a link between TOR signalling and only one of the NCL subtypes. Please consider removing “Batten disease” and replacing with “TPP1”.

Is the antisense inhibition specific for tpp1A? This is an important consideration since there are six tpp1 genes in Dictyostelium. As such, it will affect the interpretation of the data as it relates to the phenotypes that are reported. For example, are the smaller and more numerous fruiting bodies in the tpp1 knockdown line attributed to tpp1A knockdown or general tpp1 knockdown?

The authors should improve the overall appearance of the figures. The axes labels are too small to read, the boxes around the figures should be removed, the text font and size are not consistent, etc.

It is somewhat concerning that SD is not shown in most of the figures. Were these experiments repeated to validate the observed effects? The authors are urged to include error bars on their plots, especially since some of the captions state that they are, when they are not (e.g., Fig. 3).

The authors are urged to carefully proofread their paper to ensure that gene and protein nomenclature are correct. For example, Dictyostelium genes should be lowercase and italicized (e.g., tpp1). Proteins in Dictyostelium are lowercase except for the first letter, which is capitalized (e.g., Tpp1). TPP1 should only be used when referring to the mammalian protein.

Line 28-32: References are required here.

Line 46-48: Sentence should be rearranged to state “…pathological conditions including other neurodegenerative lysosomal storage disorders such as CLN3 disease, as well as inflammation, cancer and aging”.

Line 61-62: This statement about Tpp1 proteins and Dictyostelium’s phagocytic ability sort of comes out of nowhere. Please clarify.

There is a lot of text within the Introduction that is either not needed or could be cut down significantly. Please consider removing the following statements/sections:

Line 63-64

Line 68-73

Line 74-83

Line 84: “CLN genes” should be “NCL genes”.

Line 85-99: The authors should consider shortening this section. The details they choose to keep should be used to demonstrate significant findings made in Dictyostelium that validate it as a model system for studying Batten disease. I would also highlight any findings that relate back to Tpp1 function (e.g., v4-7 secretion in the cln3- cells).

Line 110-112: Revise statements to present tense (e.g., “did not” to “do not”)

Line 199-202: Are these established procedures in Dictyostelium? If so, then the references should be cited here.

Section 3.1: The authors should comment on the stability of the transformants? For example, how long does inhibition last? Does this impact any of the assays that were performed?

Line 263-266: This statement requires a citation.

Line 269: At this point, it's likely premature to conclude that the autofluorescence is due to lipofuscin accumulation since the storage material in Dictyostelium is unknown. I suggest removing lipofuscin from the text.

Line 277: Dictyostelium does not get Batten disease. I'd prefer to see the authors stick to the facts (e.g., effects of reduced expression of tpp1).

Line 279: Depending on the strain, multicellular aggregates begin to form after about 8 hours of development. "Several hours" implies that aggregate formation occurred after only 4 hours or so, which would be surprising. The authors should clarify this statement.

Line 281-283: Is counting the number of cells in a slug is a good way to quantify aggregate size? Has this been done before in Dictyostelium? What if some cells are left in the slime trail? Why not just quantify mound size? Please clarify.

Line 295: Is it correct to refer to the slug as a tissue?

Line 298: TPP1 knockdown potentially has an effect, but in this context the authors should state "To determine if Tpp1A plays a role in...".

Lien 409: Autofluorescence was recently shown in starved cln3- and cln5- cells. These results should be cited here. Also, tpp1B is highly expressed during growth. This relates back to my previous comment which questioned the specificity of the tpp1 antisense construct. Is it possible that this construct also inhibited tpp1B? Please clarify.

Line 428-431: This rationale is not clear. Just because more cells reach the point of initiating aggregation earlier, that doesn’t necessary mean they will form more aggregates. Please clarify.

Line 437: I understand what the authors are trying to say, but "accelerating" would be more appropriate here.

Line 449-450: Gomer's study saw no effect of tpp1A loss on cell proliferation. Again, I wonder if the antisense construct was inhibiting tpp1A, not tpp1B. Were the primers used in the PCR validation specific enough to only amplify tpp1A (Fig. 1)?

Line 537: Which phenotypes? Please clarify.

Line 549: “Tor” should be “TOR”.

Author Response

We would like to thank the reviewer for their time and their comments and have addressed each point below:

This study provides insight to the function of Tpp1 in Dictyostelium. Antisense inhibition was used to knockdown tpp1 expression, which affected growth, endo/phagocytosis, and development. The authors then link the knockdown phenotypes to TOR signalling.

The title of the paper should be revised since the findings suggest a link between TOR signalling and only one of the NCL subtypes. Please consider removing “Batten disease” and replacing with “TPP1”.

We have changed the title as below:

“Modelling of neuronal ceroid lipofuscinosis type 2 in Dictyostelium discoideum suggests that cytopathological outcomes result from altered TOR signalling”

Is the antisense inhibition specific for tpp1A? This is an important consideration since there are six tpp1 genes in Dictyostelium. As such, it will affect the interpretation of the data as it relates to the phenotypes that are reported. For example, are the smaller and more numerous fruiting bodies in the tpp1 knockdown line attributed to tpp1A knockdown or general tpp1 knockdown?

 Yes. The antisense inhibition fragment used does not have any significant homology with any of the other tpp1 isoforms or with any other gene. The homology between the isoforms does not exist at the DNA level only at the protein level. We have included the following sentence in the introduction to highlight this. “The six homologs show conservation at the protein but not at the DNA level.”

The authors should improve the overall appearance of the figures. The axes labels are too small to read, the boxes around the figures should be removed, the text font and size are not consistent, etc.

 We apologise for the appearance of the figures and have improved the size of the labels, removed the boxes around the figures and kept the text font size etc. consistent

It is somewhat concerning that SD is not shown in most of the figures. Were these experiments repeated to validate the observed effects? The authors are urged to include error bars on their plots, especially since some of the captions state that they are, when they are not (e.g., Fig. 3).

 Again we apologise this must have been an oversight, we have repeated the experiments several times and error bars have been added to all graphs. In some cases they are smaller than the size of the plotted point and so are not visible.

The authors are urged to carefully proofread their paper to ensure that gene and protein nomenclature are correct. For example, Dictyostelium genes should be lowercase and italicized (e.g., tpp1). Proteins in Dictyostelium are lowercase except for the first letter, which is capitalized (e.g., Tpp1). TPP1 should only be used when referring to the mammalian protein.

 We have corrected this throughout the document

Line 28-32: References are required here.

 We have added in the ref Carcel-Trullols, et al., 2015

Line 46-48: Sentence should be rearranged to state “…pathological conditions including other neurodegenerative lysosomal storage disorders such as CLN3 disease, as well as inflammation, cancer and aging”.

 We have changed the sentence accordingly

Line 61-62: This statement about Tpp1 proteins and Dictyostelium’s phagocytic ability sort of comes out of nowhere. Please clarify.

 Dictyostelium is unusual in having this many isoforms of Tpp1 and it has been suggested by Stumpf et al that this may be due to its phagocytic lifestyle. The reviewer is correct and it does not fit here so we have removed the sentence.

There is a lot of text within the Introduction that is either not needed or could be cut down significantly. Please consider removing the following statements/sections:

Line 63-64

Line 68-73

Line 74-83

 We have deleted two paragraphs from the introduction

Line 84: “CLN genes” should be “NCL genes”.

 Done

Line 85-99: The authors should consider shortening this section. The details they choose to keep should be used to demonstrate significant findings made in Dictyostelium that validate it as a model system for studying Batten disease. I would also highlight any findings that relate back to Tpp1 function (e.g., v4-7 secretion in the cln3- cells).

 We have deleted a paragraph to shorten this section.

Line 110-112: Revise statements to present tense (e.g., “did not” to “do not”)

 Done

Line 199-202: Are these established procedures in Dictyostelium? If so, then the references should be cited here.

 Have added the reference Chen et al 2017 DMM

Section 3.1: The authors should comment on the stability of the transformants? For example, how long does inhibition last? Does this impact any of the assays that were performed?

 The transformants created are stable transformants and we have added the word ‘stable’ into section 3.1. The construct integrates into the genome and is maintained by the cells.

Line 263-266: This statement requires a citation.

 There was a citation at the end of this sentence (Barth et al., [39]) but we have included the same reference for the preceding sentence to make it clearer

Line 269: At this point, it's likely premature to conclude that the autofluorescence is due to lipofuscin accumulation since the storage material in Dictyostelium is unknown. I suggest removing lipofuscin from the text.

 We have added the sentence below to clarify this:

Although the identity of the fluorescent material is not known, this is consistent with an accumulation of lipofuscin as in mammalian cells”.

Line 277: Dictyostelium does not get Batten disease. I'd prefer to see the authors stick to the facts (e.g., effects of reduced expression of tpp1).

 We have changed the text accordingly

Line 279: Depending on the strain, multicellular aggregates begin to form after about 8 hours of development. "Several hours" implies that aggregate formation occurred after only 4 hours or so, which would be surprising. The authors should clarify this statement.

 Yes the reviewer is correct. We observed the development after 12 hours and whilst the AX2 parental strain only displayed mounds more development had occurred in the Tpp1 strains indicating faster progression through development. We have changed the sentence to “With regard to morphology, we observed that Dictyostelium Tpp1 mutants developed into  multicellular aggregates quicker than wild type strains”

Line 281-283: Is counting the number of cells in a slug is a good way to quantify aggregate size? Has this been done before in Dictyostelium? What if some cells are left in the slime trail? Why not just quantify mound size? Please clarify.

In general the amount of cells that are left behind in the slug trail is minimal especially in Tpp1 mutants which produce small slugs which do not migrate very far. We have however changed the text so that we only refer to slug size and not to aggregate size as shown below:

“Because of the smaller fruiting body sizes we also examined if the mutants formed smaller slugs. This was confirmed quantitatively by counting cell numbers per slug for each transformant, the results revealing a copy number-dependent decrease in slug size.”

Line 295: Is it correct to refer to the slug as a tissue?

 We have changed the word “tissues” to “cells”

Line 298: TPP1 knockdown potentially has an effect, but in this context the authors should state "To determine if Tpp1A plays a role in...".

 Changed

Lien 409: Autofluorescence was recently shown in starved cln3- and cln5- cells. These results should be cited here. Also, tpp1B is highly expressed during growth. This relates back to my previous comment which questioned the specificity of the tpp1 antisense construct. Is it possible that this construct also inhibited tpp1B? Please clarify.

We have added in the text:

“Autofluorescence has been observed in another Dictyostelium NCL model the CLN5 null mutant [25].”

We could not find the reference for observed autofluorescence in a CLN3 null strain. But if the reviewer could provide this reference we would happily include it. As mentioned in an earlier comment the conservation between the tpp1 isoforms is at the protein level and not at the DNA level. The DNA fragment of tpp1A did not have homology with any other gene including other tpp1 genes.

Line 428-431: This rationale is not clear. Just because more cells reach the point of initiating aggregation earlier, that doesn’t necessary mean they will form more aggregates. Please clarify.

 Aggregation centers are formed due to cells which differentiate to the point of being excitable and able to begin emitting cAMP pulses. If more cells develop to this point at around the same time (as often happens with accelerated development), then there will be more aggregation centres and thus more aggregates. The overall numbers of cells is the same and the aggregate sizes are therefore smaller.  To avoid any confusion we have removed the word “earlier” after “initiating aggregation”.

Line 437: I understand what the authors are trying to say, but "accelerating" would be more appropriate here.

 We have changed the sentence as below to clarify:

It is not surprising that Tpp1 plays a role in suppressing the initiation of normal development so that its loss accelerates development.”

Line 449-450: Gomer's study saw no effect of tpp1A loss on cell proliferation. Again, I wonder if the antisense construct was inhibiting tpp1A, not tpp1B. Were the primers used in the PCR validation specific enough to only amplify tpp1A (Fig. 1)?

 Yes the primers used in PCR validation were specific for tpp1A and no other gene including any tpp1 isoforms

Line 537: Which phenotypes? Please clarify.

 We have adjusted the sentence as below:

“Overexpression of Rheb rescued the slug size, growth and endocytic phenotypes caused by antisense inhibition of Tpp1”.

Line 549: “Tor” should be “TOR”.

Done

Reviewer 4 Report

In the paper Stumpf et al. shows a correlation between Tpp1 and TOR signaling in Dictyostelium. Knockdowns of TPP1 by multiple integrations of an antisense construct was shown to phenocopy prolonged rapamycin treatment or Rheb knockdown, while the overexpression of Rheb rescued the phenotype.

Although the authors show a correlation between Tpp1 antisense copy numbers and an array of phenotypes, they do not show that the phenotypes could be complemented (rescued) by Tpp1 expression itself (i.e. overexpression). This should be shown in order to demonstrate that the resulting phenotypes are fully TPP1-specific, as integration of  antisense construct was random and in a high copy number. Western analysis should also be used to demonstrate that the reduction  in Tpp1 mRNA expression actually leads to reduced Tpp1 protein levels. This would be important to show as well.

The Tpp1 knockdown phenotype was rescued by the overexpression of Rheb and was phenocopied by prolonged exposure to rapamycin. Although the authors show this correlation between Tpp1 knockdown and TOR signaling, they should show that TOR signaling is indeed downregulated in TPP1 knockdown/mutant. The author should test if the effect is due to TORC1 or TORC2 by a more direct means – i.e. knockdown of TORC-specific components. Can the authors show altered TPP1 protein localization upon rapamycin treatment, Rheb knockdown? Is the interaction of TPP1 with GPHR affected by rapamycin treatment?

In addition to reference 56, which demonstrated a potential connection between TOR signaling and CLN3 (BTN1) mutant phenotypes in yeast, another (earlier) paper described the direct connection between TOR signaling and defects in Golgi quality control which phenocopy the mutation of CLN3/BTN1 in yeast (Dobzinski et al 2015, Cell Rep 12:1876). Since the authors have already established a connection between TPP1 and Golgi resident proteins (Stump et al 2017, Dis Model Mech) it would merit some discussion in regards to how the authors believe TPP1 function or localization is actually regulated by TOR signaling.

Other points:

1)     Figure 3 – Change the black font on the picture to white

2)     Figure 7 – There is an overly of the graphs in the pdf - rearrange the figure

Author Response

We would like to thank the reviewer for their time and their comments and have addressed each point below:

In the paper Stumpf et al. shows a correlation between Tpp1 and TOR signaling in Dictyostelium. Knockdowns of TPP1 by multiple integrations of an antisense construct was shown to phenocopy prolonged rapamycin treatment or Rheb knockdown, while the overexpression of Rheb rescued the phenotype.

Although the authors show a correlation between Tpp1 antisense copy numbers and an array of phenotypes, they do not show that the phenotypes could be complemented (rescued) by Tpp1 expression itself (i.e. overexpression). This should be shown in order to demonstrate that the resulting phenotypes are fully TPP1-specific, as integration of  antisense construct was random and in a high copy number. Western analysis should also be used to demonstrate that the reduction  in Tpp1 mRNA expression actually leads to reduced Tpp1 protein levels. This would be important to show as well.

 To overexpress Tpp1 at the same time as antisense inhibition of Tpp1 is essentially just producing strains with different levels of Tpp1. This is what we have done by using multiple transformants each with different levels of Tpp1 mRNA. We also tried to overexpress Tpp1 but could not isolate transformants. We agree with the reviewer that western analysis would be good but we do not have a Tpp1 antibody to do this and could find no commercial antibody that works.

The Tpp1 knockdown phenotype was rescued by the overexpression of Rheb and was phenocopied by prolonged exposure to rapamycin. Although the authors show this correlation between Tpp1 knockdown and TOR signaling, they should show that TOR signaling is indeed downregulated in TPP1 knockdown/mutant. The author should test if the effect is due to TORC1 or TORC2 by a more direct means – i.e. knockdown of TORC-specific components. Can the authors show altered TPP1 protein localization upon rapamycin treatment, Rheb knockdown? Is the interaction of TPP1 with GPHR affected by rapamycin treatment?

We attempted to show that TOR signalling was affected by using antibodies directed at mammalian TORC1 substrates but these did not work in Dictyostelium in our hands. To knock down TORC-specific components would involve the creation of many more strains and their full phenotypic analysis which is beyond the scope of this publication. We do not have an antibody against Tpp1 so cannot show altered localisation. The interaction mentioned between Tpp1 and GPHR was described by other authors for a different Tpp1 isoform (Tpp1F not Tpp1A).

In addition to reference 56, which demonstrated a potential connection between TOR signaling and CLN3 (BTN1) mutant phenotypes in yeast, another (earlier) paper described the direct connection between TOR signaling and defects in Golgi quality control which phenocopy the mutation of CLN3/BTN1 in yeast (Dobzinski et al 2015, Cell Rep 12:1876). Since the authors have already established a connection between TPP1 and Golgi resident proteins (Stump et al 2017, Dis Model Mech) it would merit some discussion in regards to how the authors believe TPP1 function or localization is actually regulated by TOR signaling.

We have not made any mention of Golgi quality control and have not made any claims about TOR regulating Tpp1A function or localisation. The reviewer states that we have already established a connection, however that work was done by a different group (Stumpf et al.) and was in relation to Tpp1F not Tpp1A.

Round 2

Reviewer 1 Report

My comments have been addressed apart from an experiment which would be outside timescales.

Author Response

Again we would like to thank the reviewer for their time

Reviewer 2 Report

My responses to the authors' notes are below (in blue).

We would like to thank the reviewer for their time and comments and have addressed each of the points below:

In this study, the authors generate Tpp1A antisense transformants in Dictyostelium and show: enhanced cellular autofluorescence; enhanced endocytic activity (phagocytosis and pinocytosis); slowed vegetative growth; and developmental defects (precocious aggregation and smaller fruiting bodies).  They further show that TOR inhibition (via rapamycin, or rheb antisense transformants) in several ways phenocopies Tpp1A antisense transformants.  Thus the authors suggest that Tpp1A has a role in TOR signaling pathways.

Several pieces of data are strong, showing significant and interesting effects (quite striking effect on fruiting body size).  However, there are open questions that require resolution before this study would be acceptable for publication.

A major concern is poor data correlating Tpp1A transcript abundance with antisense copy number. Only vegetative cells were examined for Tpp1a transcript abundance, and the results as presented are difficult to interpret.  Since it has been reported that TPP1A expression is low/absent during vegetative growth, while it is induced with development, peaking at 16h (Stumpf et al. 2017), it would be logical to assess transcript abundance at developmental stages.  Are RT-PCR primers outside of the 1361 bp antisense sequence, to avoid possible amplification of antisense transcripts?

The reviewer is correct that the transcript of Tpp1A is reported to be very low in the vegetative stage, yet our results show that despite this low expression there are phenotypic effects of further reducing Tpp1A expression. For this reason, and because many of the phenotypes that we measured were done using vegetative cells, we thought it appropriate (and technically easier) to measure RNA levels in this stage. We were able measure RNA levels at the vegetative stage (thus it is not zero) and we could see a significant 32-64 fold reduction in Tpp1 expression.

One of your clones shows about 11 cycle difference – this would be about 2000-fold reduction. How do you measure 2000-fold reduction from already very low-level expression? I am wondering if the qPCR is sensitive enough to reliably detect copy number-dependent drop in transcript abundance in vegetative state samples? Please include in the text of the manuscript the mean Ct value for the WT, to give the reader a reference.  It would be appropriate to test both vegetative and an early development time-point for transcript levels (phenotypes are shown for both), at least for a few representative transformants, for more convincing evidence of dose-dependent antisense knock-down.  

Again, the reviewer is correct and RT-PCR primers could, in principle, amplify antisense transcripts. The RT stage of the experiment had only the reverse primer so only the mRNA could be reverse transcribed. The RT itself is not thermostable, so is destroyed in the PCR stage of the experiment. However, low levels of reverse transcriptase activity in the thermostable DNA polymerase in the PCR step can occasionally result in the antisense transcript also being amplified, as the reviewer suggests.  In our experience with many genes and many primer sets, this problem can occur occasionally but is unusual and evidently did not occur here. If it had then we would expect to see increased qRT-PCR signal in the Tpp1 antisense transformants rather than the observed decrease.

I have encountered RT to occur in the absence of any added primer (thought due to RNA fragments acting as primers), when performing two-step RT-qPCR.  This can be minimized with type of RT, temperature, etc.  But as you point out, this is not likely a concern here.

Additional comments/concerns are listed below.

1) What is the promoter in the pDneo2 vector, and is it active in vegetative and developmental stages? What is the HPF261 line? It has 35 copies of what?

 The pDNeo2 promoter controlling the expression of the inserted gene is the Actin 6 promoter. A paper by Knecht et al., (1986) Mol. Cell. Biol. shows that expression of proteins from this promoter results in high expression at 0-6 hours of development and this is reduced down to low levels (but not to zero) at 20 hours of development. This indicates that the antisense mRNA would be expressed more strongly during the vegetative and early stages of development. In this case that may help with successful antisense inhibition of the lower concentrations of the target mRNA by increasing the concentrations of the inhibitor mRNA in vegetative cells compared to cells in the multicellular stages. At the later stages the high copy numbers of the construct will also offset any downregulation of the promoter in the construct. The fact that we see phenotypes at both stages indicates that the knock down is effective at both stages.

I asked about the promoter and HPF261 as an indicator that such information should be included in the manuscript (in a brief fashion).

HPF261 is a control strain which has a known number of integrated constructs (35 copies), it is used to normalise the data. We have added “of a control construct – the parental vector into which the antisense fragments were cloned.” to clarify this.

Thank you.

2) Might the 1361 bp antisense have enough sequence similarity with homologs to reduce their expression levels? RT-qPCR should be performed for the other TPP1 homologs in addition to TPP1A, with RNA samples collected from vegetative and developmental stages. Phenotypes at vegetative stages could be due to “off target” knock-down of a homolog.

 The Tpp1A sequence and the fragment used for antisense inhibition show no sequence homology at the DNA level with any other gene including all of the Tpp1 isoforms. The conservation between the Tpp1 isoforms is at the protein level and not at the DNA level.

There are in fact small stretches of cDNA sequence identity between Tpp1a and Tpp1b and Tpp1e, (similar to length of a miRNA, at about 20-25bp).  Given that Tpp1b is far more abundant during the vegetative state than Tpp1a, I stand by my opinion that it is important to rule out the possible down-modulation of Tpp1b in contributing to the observed phenotypes, particularly veg state phenotypes.  It may be an off-chance, but would serve as a relevant negative control for off-target effect.  Again, this could be done with a few select transformants vs WT.

3) Why do the authors describe their qRT-PCR as semiquantitative, when it is quantitative? (provides relative or absolute quantitative data).

Because “relative quantitative data” would refer to quantitative data for one gene that has been expressed relative to quantitative data for another gene. We did not use calibration curves to determine quantitative expression levels for the internal control housekeeping gene (filamin) or for the gene of interest. The expression levels are thus measured from differences in Ct cycle numbers between the test and the control gene. These are therefore “semiquantitative” not “relative quantitative”.

I believe that “semi-quantitative” refers to results from “conventional PCR” (not real-time PCR), wherein PCR products are run on a gel and the band densities compared, as a crude measure. Real-time PCR (aka qPCR) provides quantitative data, regardless of whether the data are expressed as absolute (using calibration curve) or relative (as you report, using “delta delta Ct” analysis). But I also now realize that opinions are mixed on this usage, and there are other papers consistent with yours.

4) Figure 1 is meant to assess a relationship between antisense copy number and Tpp1A transcript abundance, but the manner of displaying this data seems odd.  Why not simply plot “antisense copy number” (as positive numbers, not negative numbers) on the x axis and relative transcript abundance on the y axis? The use of “expression index” as a label is confusing if one is not aware of “the convention”, particularly when “expression” is on the Y axis.  Also, relative transcript abundance should be plotted relative to WT, where WT = “1” (or 100%), for ease of understanding.

The number of antisense copy numbers is plotted as a negative number because further in the paper we plot the Rheb antisense construct copy number and Rheb overexpression copy number on the same graph. The negative numbers correspond to antisense inhibition as it is reducing expression and the positive numbers to overexpression as it is increasing Tpp1 levels. For consistency and to follow the convention used in this kind of work established by Bokko et al. (2007), we have kept all antisense construct numbers as negative numbers. This way the expression levels always increase as one moves from left to right on the X axis. In the case of Figure 1 we have changed the X axis title to construct copy number as suggested by the reviewers. The AX2 control value is plotted at 0 because qPCR cycle numbers are a logarithm of the increase in the amount of the template (which grows exponentially). So the correct way to normalize against the control is not by division but by subtraction (the logarithmic equivalent of division). A value of zero means no change and is the logarithm of 1 (which would be the value if the cycle numbers had been converted to template quantities and division used to normalize).

I understand the reasoning behind the convention, but still disagree with its use.  It is misleading to label an axis as “expression index” when it is in fact copy number. Your own Figure 1, indicates that expression level does NOT always increase as one moves from left to right on the x axis.  Why not label the axis as “antisense copy number” to the left of 0, and “sense copy number” to the right of 0? (example L to R from 200 down to 0, then up to 200).

I have no concern about the method of normalization.  My point was that “relative transcript abundance” is typically graphed as a proportion of the (untreated) control, rather than as change in Ct. For example, a 3 Ct difference is an 8-fold decrease, and would be graphed as 12.8%, with the non-transformed control at 100%. That said, your plot of Ct is an acceptable alternative.

5) Lipofuscin is insoluble aggregates of oxidized proteins, lipids and carbohydrates, and should show broad ex/em fluorescence, under blue as well as UV excitation. Autofluorescence in cells does not necessarily mean lipofuscin.  Increased cellular autofluorescence might merely reflect metabolic shift, with increased levels of naturally fluorescent molecules.  Do the Rheb anti-sense transformants similarly show increased autofluorescence?

The published ranges for excitation emission spectra for lipofuscin in Batten disease cells are350-380nm and 400-600 nm respectively (Seehafer and Pearce 2006 [45]). We did not observe a difference in the Blue wavelength in agreement with Huber and Mathavarajah, 2018, who also observed autofluorescence in the CLN5 null mutant with excitation: 340–380 nm, emission: 440–480 nm but not with excitation: 475–495 nm, emission: 565–615 nm.

From the Seehafer and Pearce review: “lipofuscin has been described by its spectral properties, with an excitation at between 320 and 480 nm and an emission wavelength between 460 and 630 nm”.  This is consistent with the 1981 Dowson microscopic study that used ex/em 390-490/>515 to capture the strongest lipofuscin signal in brain section neurons.  From various sources it seems that lipofuscin, in Batten disease or with aging, has very broad ex/em.  In fact, the product “Trueblack” was developed to quench lipofuscin autofluorescence across all standard channels.  If present, lipofuscin could certainly contribute to fluorometer readings (365/410-460), but I would bet that with proliferating cells, most/all of this signal is from non-lipofuscin, metabolism-related fluorescent molecules (such as NADH/NADPH).  It is plausible that Tpp1 knock-down could influence metabolism.

We agree with the reviewers that autofluorescence in these cells does not necessarily mean lipofuscin and have changed the sentence to clarify this as shown below:

“Although the identity of this fluorescent material is not known, this is consistent with an accumulation of lipofuscin as in mammalian cells”.

Thank you

We have not measured autofluorescence in the Rheb antisense transformants.

Is the autofluorescence intensified after change to developmental (nutrient poor) media? Perhaps the authors could stain for ATP synthase subunit c (Neufeld Ab to detect the aggregated form). As the major protein component of Batten lipofuscin, this would be a better disease correlate.  ATP synthase subunit c aggregates should intensify over time during post-mitotic development, in absence/knock-down of TPP1A.

The antibody described by the reviewer is developed against the mouse protein. The sequence used for development of this antibody when used in a BLAST search against the Dictyostelium protein database does not detect any similar proteins.

Thought it was worth to suggest. Unfortunate that the protein is not conserved, as the immuno-stain is a wonderful marker.

6) Rather than co-transform, could the authors have performed sequential transformation, for more a direct assessment of phenotypic rescue?

Unfortunately this is not possible using the constructs we have, as both contain the same selectable marker.

I assumed that was the case.  Would entail cloning in a different selectable marker to the second construct. Thought for a future study.

7) The statement “the only experimental evidence for the role of TOR signaling in Batten disease came from studies in yeast…”, is not accurate.  Several CLN3 and CLN2 studies show effects on TOR signaling pathways, in relation to autophagy and/or signaling complex formation at the lysosome (eg. from Cotman, Sardiello labs; and a 2013 paper by Vidal-Donet et al.).

 We apologise for this oversight and have removed the sentences which indicated this. The final paragraph of the discussion has been changed as below:

“TOR signalling has previously been suggested to play a role in Batten disease. Our results agree with studies in yeast investigating the CLN3 homolog Btn [56]. Synthetic genetic arrays identified interactions between Btn and the core components of TORC1 and TORC2. The Btn null cells exhibited phenotypes consistent with defective TOR signalling [56]. Our results similarly support a role of TOR signalling in Batten disease, in this case the form associated with loss of CLN2 function.”

8) the title should specify “CLN2 Batten disease”.

We have changed the title as below:

“Modelling of neuronal ceroid lipofuscinosis type 2 in Dictyostelium discoideum suggests that cytopathological outcomes result from altered TOR signalling”

Thanks.

Author Response

We would again like to thank the reviewer for their time and comments and have addressed each comment as below:

One of your clones shows about 11 cycle difference – this would be about 2000-fold reduction. How do you measure 2000-fold reduction from already very low-level expression? I am wondering if the qPCR is sensitive enough to reliably detect copy number-dependent drop in transcript abundance in vegetative state samples? Please include in the text of the manuscript the mean Ct value for the WT, to give the reader a reference.  It would be appropriate to test bothvegetative and an early development time-point for transcript levels (phenotypes are shown for both), at least for a few representative transformants, for more convincing evidence of dose-dependent antisense knock-down.  

The reviewer is correct that the sensitivity levels of the qRT-PCR are approaching their limits in the highly antisense inhibited strains. Yet the Ct values for those strains are still within the limits of detection being under 28 with our negative controls having Ct values around 35. We have included the following in Figure 1 legend: “Prior to normalization the Ct value for amplification of the tpp1 and filamin gene using the wild type AX2 strain was 21.”

As suggested by the reviewer we starved four of the Tpp1 antisense inhibited strains and the wild type cells for 8 hours, extracted the RNA and repeated the qRT-PCR. We did see a reduction in tpp1A mRNA in aggregation competent cells and also in vegetative cells. We have changed Figure 1 to include this data. Whilst we were doing these experiments we also retested the strain which showed the most inhibition as this strain showed a lot of variation in the measurements. This retesting showed that the inhibition was actually 7 fold rather than the initially described 11 fold reduction. This has also been adjusted in Figure 1.

I asked about the promoter and HPF261 as an indicator that such information should be included in the manuscript (in a brief fashion).

We have added “under the control of the Actin-6 promoter” in the following sentence. The fragment was cloned into the EcoRI and BamHI sites of the pDNeo2 vector in the antisense orientation under the control of the Actin-6 promoter.

We had also included the following sentence to explain the HPF261 strain: “control strain HPF261 which is known to have 35 copies of a control construct, the parental vector into which the antisense fragments were cloned.”

There are in fact small stretches of cDNA sequence identity between Tpp1a and Tpp1b and Tpp1e, (similar to length of a miRNA, at about 20-25bp).  Given that Tpp1b is far more abundant during the vegetative state than Tpp1a, I stand by my opinion that it is important to rule out the possible down-modulation of Tpp1b in contributing to the observed phenotypes, particularly veg state phenotypes.  It may be an off-chance, but would serve as a relevant negative control for off-target effect.  Again, this could be done with a few select transformants vs WT.

The Tpp1A sequence used for antisense inhibition is from 385-1745bp. In contrast to mammalian cells antisense inhibition in Dictyostelium is much more efficient using larger fragments, the size range commonly used is around 500bp. This is based on studies where smaller fragments did not result in reduction of the target mRNA. An example is a study done to antisense inhibit the myosin heavy chain, fragments of 3.7 and 1.5 kb effectively reduced the expression of this gene however a 396bp fragment was not effective. It was also noted in this study that the myosin isoform 1B which shares 57% identity with the target gene had no reduction in RNA levels indicating that the antisense inhibition was specific despite the sequence similarities. There are also reports of this in other models such as Drosophila.

Nonetheless to investigate the possibility of off-target knockdown of tpp1B or tpp1E, we performed BLAST alignments of their DNA sequences with the sequence of our antisense RNA fragment. The reviewer is correct in that there was a single, very short (11bp) stretch of tpp1A sequence, 9bp of which were identical with sequences in tpp1B (1735-1743 of tpp1A) or tpp1E (1737-1745 of tpp1A). To determine if a 9 bp stretch of sequence complementarity between our tpp1A antisense RNA and the mRNAs of other genes could cause significant off-target knockdown, we used qRT-PCR to measure tpp1B mRNA expression levels in several of our tpp1A knockdown strains. The tpp1B mRNA levels were not significantly reduced - relative Ct=-0.5+1.7 (95% confidence interval), not significantly different from zero. This shows that 9 bp of sequence identity is insufficient to produce any significant knockdown of expression.

It is also possible that altered tpp1A expression may result in alterations in the regulation of other tpp1 isoforms. We mentioned in the paper that Tpp1F contained Tpp1 enzyme activity yet the knockout strain displayed no abrogation of total Tpp1 activity, presumably due to compensation by the other homologs. Whether or not Tpp1A knockdown causes dysregulation of other Tpp1 isoforms is beyond the scope of our manuscript and has no impact on our conclusion that the downstream consequences are mediated by TOR complex signalling.

I understand the reasoning behind the convention, but still disagree with its use.  It is misleading to label an axis as “expression index” when it is in fact copy number. Your own Figure 1, indicates that expression level does NOT always increase as one moves from left to right on the x axis.  Why not label the axis as “antisense copy number” to the left of 0, and “sense copy number” to the right of 0?[PF3]  (example L to R from 200 down to 0, then up to 200[SA4] [PF5] ).

We have changed the labelling of the axes in all figures to construct copy number rather than expression index

I have no concern about the method of normalization.  My point was that “relative transcript abundance” is typically graphed as a proportion of the (untreated) control, rather than as change in Ct. For example, a 3 Ct difference is an 8-fold decrease, and would be graphed as 12.8%, with the non-transformed control at 100%. That said, your plot of Ct is an acceptable alternative[PF6] .

We understand that this is often the way that qRT-PCR levels are expressed. We chose not to do so, because the conversion of Ct differences to fold-changes involves an assumption about the efficiency of the amplification process – ie a cycle number difference of 1 is assumed to represent a 2-fold difference. This need not actually be the case for a variety of reasons, so the Ct difference representation avoids making this assumption.

From the Seehafer and Pearce review: “lipofuscin has been described by its spectral properties, with an excitation at between 320 and 480 nm and an emission wavelength between 460 and 630 nm”.  This is consistent with the 1981 Dowson microscopic study that used ex/em 390-490/>515 to capture the strongest lipofuscin signal in brain section neurons.  From various sources it seems that lipofuscin, in Batten disease or with aging, has very broad ex/em.  In fact, the product “Trueblack” was developed to quench lipofuscin autofluorescence across all standard channels.  If present, lipofuscin could certainly contribute to fluorometer readings (365/410-460), but I would bet that with proliferating cells, most/all of this signal is from non-lipofuscin, metabolism-related fluorescent molecules (such as NADH/NADPH).  It is plausible that Tpp1 knock-down could influence metabolism.

Yes, it is possible that Tpp1A knockdown could influence metabolism and we are planning future experiments to measure this. At this stage though we do not have any data to support this.

Reviewer 3 Report

Line 29: “NCL’s” should be replaced with “NCLs”.

Line 56: I would provide some discussion about Dictyostelium prior to elaborating on its use as a model system for NCL.

Line 73: This is not clear. I understand that "not at the DNA level" was added to address one of the reviewer comments, but I think it ends up confusing the reader at this point. It would be better to address the reviewer comment in the methods when the antisense construct is discussed.

Line 90: I'd insert "obvious defects". There may be subtle defects that have not yet been identified.

Line 103: As mentioned in the previous review, the authors are urged to carefully proofread the text. Tpp1A should be /tpp1A/. This error appears numerous times in the text.

Line 235: How can the antisense inhibition integrate into random sites in the genome if it is targeted against a specific region? Please clarify.

Line 320: “R” missing from “rapamycin”

Line 393: Delete this statement since it is present below.

Line 420: Insert “also” after “autofluorescence has…”.

Author Response

We would like to thank the reviewer again for their time and their comments and have addressed each comment as below:

Line 29: “NCL’s” should be replaced with “NCLs”.

Fixed

Line 56: I would provide some discussion about Dictyostelium prior to elaborating on its use as a model system for NCL.

We have included the following paragraph in the introduction:

Dictyostelium discoideum is one of these models. Dictyostelium is a cellular slime mold or social amoeba. It has all the benefits of a model system with a haploid, completely sequenced genome. It is genetically tractable and is amenable to a range of biochemical, cell biological and physiological studies [38]. In addition, Dictyostelium has a unique life cycle with unicellular and multicellular stages. It begins the life cycle as a unicellular amoebae feeding on microorganisms in the soil. When the food source is depleted the cells begin to emit and respond to a chemical signal, cAMP. This leads to the formation of an aggregate consisting of approximately 105 cells which undergoes multicellular development through multiple stages leading to the final structure of a fruiting body consisting of a basal disc a long slender stalk containing cells which have undergone autophagy and a sorus containing spores. This unique life cycle provides a plethora of phenotypes for study which are essentially readouts of the underlying signalling pathways [38].

Line 73: This is not clear. I understand that "not at the DNA level" was added to address one of the reviewer comments, but I think it ends up confusing the reader at this point. It would be better to address the reviewer comment in the methods when the antisense construct is discussed.

We have deleted the sentence in the introduction and added the following sentence in the methods section:

This DNA fragment contains no significant sequence similarity with any of the other tpp1 genes.

Line 90: I'd insert "obvious defects". There may be subtle defects that have not yet been identified.

Done

Line 103: As mentioned in the previous review, the authors are urged to carefully proofread the text. Tpp1A should be /tpp1A/. This error appears numerous times in the text.

Line 235: How can the antisense inhibition integrate into random sites in the genome if it is targeted against a specific region? Please clarify.

The construct inserts randomly into the genome through creation of a single nick in the genome. Targeted homologous recombination could also occur yet the frequency of this is much lower, in most reports being below 1 in 100. Antisense inhibition acts at the RNA level, while targeted integration by homologous recombination acts at the DNA level. These are two different processes that use different enzymatic machinery and operate at very different levels of efficiency.

Line 320: “R” missing from “rapamycin”

Fixed

Line 393: Delete this statement since it is present below.

We apologise for the error and have deleted the sentence

Line 420: Insert “also” after “autofluorescence has…”.

We have changed the sentence: “Autofluorescence has also been observed in Dictyostelium CLN5 null mutants”

Reviewer 4 Report

Apologies for the error in author citation. I understand that a specific antibody is not available for use with TPP1a, but can't the authors use genome tagging or plasmid-based expression to introduce an epitope-tagged form for Western analysis (to verify and quantify changes in protein as a consequence of antisense expression) or a fluorescent protein-tagged form to examine TPP1 localization? The latter might be revealing as to the functionality of the protein upon rapamycin treatment, etc. 

Author Response

We thank the reviewer for their time and have addressed their comments as below:

Apologies for the error in author citation. I understand that a specific antibody is not available for use with TPP1a, but can't the authors use genome tagging or plasmid-based expression to introduce an epitope-tagged form for Western analysis (to verify and quantify changes in protein as a consequence of antisense expression) or a fluorescent protein-tagged form to examine TPP1 localization? The latter might be revealing as to the functionality of the protein upon rapamycin treatment, etc

Yes the reviewer is correct an epitope-tagged Tpp1A could be created and it could be used for localisation studies. It would not however be useful for quantifying changes in the antisense strains as by expressing the tagged protein we would be overexpressing it and hence the strains would not have reduced Tpp1A expression anymore. In addition we have tried to isolate Tpp1A overexpression strains with no success suggesting it may be detrimental to the cells. In regards to rapamycin expression we believe that rapamycin is working downstream of Tpp1a on TORC1 and hence rapamycin treatment may not have any effect on the function of Tpp1a. Although creation of an epitope-tagged Tpp1A protein could be beneficial for future studies it is beyond the scope of this paper.